# Learning Social Welfare Functions

**Kanad Shrikar Pardeshi**
Carnegie Mellon University
kpardesh@andrew.cmu.edu

**Itai Shapira**
Harvard University
itaishapira@g.harvard.edu

**Ariel D. Procaccia**
Harvard University
arielpro@seas.harvard.edu

**Aarti Singh**
Carnegie Mellon University
aarti@andrew.cmu.edu

## Abstract

Is it possible to understand or imitate a policy maker's rationale by looking at past decisions they made? We formalize this question as the problem of learning social welfare functions belonging to the well-studied family of power mean functions. We focus on two learning tasks; in the first, the input is vectors of utilities of an action (decision or policy) for individuals in a group and their associated social welfare as judged by a policy maker, whereas in the second, the input is pairwise comparisons between the welfares associated with a given pair of utility vectors. We show that power mean functions are learnable with polynomial sample complexity in both cases, even if the social welfare information is noisy. Finally, we design practical algorithms for these tasks and evaluate their performance.

## 1 Introduction

Consider a standard decision making setting that includes a set of possible actions (decisions or policies), and a set of individuals who assign utilities to the actions. A *social welfare function* aggregates the utilities into a single number, providing a measure for the evaluation of actions with respect to the entire group. Utilitarian social welfare, for example, is the sum of utilities, whereas egalitarian social welfare is the minimum utility. Given two actions that induce the utility vectors $(3, 0)$ and $(1, 1)$ for two individuals, the former is preferred when measured by utilitarian social welfare, whereas the latter is preferred according to egalitarian social welfare.

When competent decision makers adopt policies that affect groups or even entire societies, they may have a social welfare function in mind, but it is typically implicit. Our goal is to *learn* a social welfare function that is consistent with the decision maker's rationale. This learned social welfare function has at least two compelling applications: first, *understanding* the decision maker's priorities and ideas of fairness, and second, potentially *imitating* a successful decision maker's policy choices in future dilemmas or in other domains.

As a motivating example, consider the thousands of decisions made by public health officials in the United States during the Covid-19 pandemic: opening and closing schools, restaurants, and gyms, requirements for masking and social distancing, lockdown recommendations, and so on. Each decision induces utilities for individuals in the population; closing schools, for instance, provides higher utility to medically vulnerable individuals compared to opening them, but arguably has much lower utility for students and parents. Assuming that healthcare officials were acting in the public interest and (approximately) optimizing a social welfare function, which one did they have in mind? Our goal is to answer such questions by learning from example decisions.

Another example we consider in this paper is that of allocating food resources in a community by a US-based nonprofit to hundreds of recipient organizations. Working with a dataset of utility of 18

Table 1: A summary of our results regarding the sample complexity of various tasks. Here, $\xi = u_{\max}(u_{\max} - u_{\min})$ and $\kappa = \log(u_{\max}/u_{\min})$, with all $d$ individual utilities assumed to be in the range $[u_{\min}, u_{\max}]$. $\rho \in [0, 1/2)$ is the probability of mislabeling for the i.i.d noise model, and $\tau_{\max}$ is the maximum temperature of the logistic noise model.

| Social Welfare Information | Loss | Known Weights | Unknown Weights |
|---|---|---|---|
| Cardinal values | $\ell_2$ | $\mathcal{O}(\xi^2)$ | $\mathcal{O}(\xi^2 d \log d)$ |
| Pairwise comparisons | 0-1 | $\mathcal{O}(\log d)$ | $\mathcal{O}(d \log d)$ |
| Pairwise comparison with i.i.d noise | 0-1 | $\mathcal{O}\left(\frac{\log d}{(1-2\rho)^2}\right)$ | $\mathcal{O}\left(\frac{d \log d}{(1-2\rho)^2}\right)$ |
| Pairwise comparisons with logistic noise estimation | Logistic | $\mathcal{O}(\tau_{\max}^2 \kappa^2)$ | $\mathcal{O}(\tau_{\max}^2 \kappa^2 d \log d)$ |

different stakeholders such as donors, volunteers, dispatchers and recipient organizations [11], we consider the task of learning the social welfare implicit in the decisions that may be made by the nonprofit.

In order to formalize this problem, there are two issues we need to address. First, to facilitate sample-efficient learnability, we need to make some structural assumptions on the class of social welfare functions. We focus on the class of *weighted power mean functions*, which includes the most prominent social welfare functions: the aforementioned utilitarian and egalitarian welfare, as well as Nash welfare (the product of utilities). This class is a natural choice, as it is the only class of functions feasible under a set of reasonable social choice axioms such as monotonicity, symmetry, and scale invariance [20, 7].

Second, we need to specify the input to our learning problem. There are two natural options, and we explore both: utility vectors coupled with their values under a target social welfare function, or pairwise comparisons between utility vectors. We demonstrate sample complexity bounds for both types of inputs, where the social welfare value or comparisons can be noiseless or corrupted by noise. We note that estimating the utility vector associated with any particular decision or policy is ostensibly challenging, but in fact this has been done in prior work and we have access to relevant data, as we discuss in Section 6.

**Our contributions.** Learning weighted power mean functions is a non-standard regression or classification problem due to the complex, highly nonlinear dependence on the power parameter $p$, which is the parameter of interest. While one can invoke standard hyperparameter selection approaches such as cross-validation to select $p$ from a grid of values, the infinite domain of $p$ does not allow demonstration of a polynomial sample complexity without deriving an appropriate cover. We derive statistical complexity measures such as pseudo-dimension, covering number, VC dimension and Rademacher complexity for this function class, under both cardinal and ordinal observations of the social welfare function. Our sample complexity bounds are summarized in Table 1. These results may be of interest for other problems where weighted power mean functions are used, such as fairness in federated learning [12].

We highlight some key contributions of this paper. We first establish the statistical learnability of widely used social welfare functions belonging to the weighted power mean functions family. We derive a polynomial sample complexity of $\mathcal{O}(1)$ for learning using cardinal social welfare values under $\ell_2$ loss, and $\mathcal{O}(\log d)$ (where $d$ denotes the number of individuals) for learning using comparisons under $0-1$ loss in the unweighted/known weight setting. The upper bounds leverage the monotonicity of the target functions with $p$ in the cardinal case, and the restricted number of roots in the ordinal setting. We also prove matching lower bounds for the ordinal case.

We also establish a polynomial sample complexity of $\mathcal{O}(d \log d)$ for both cardinal and ordinal tasks in the setting when the individual weights are unknown. This result is intuitive, as learning an additional $d$ weight parameters incurs a proportional increase in the sample requirement.

We then analyze the sample complexity for the more practical ordinal task under different noise models (i.i.d. and logistic noise) and characterize the effect of noise on learning. In the i.i.d. noise setting, sample complexity increases with noise level $\rho$, converging to the noiseless complexity as $\rho \to 0$. Unlike the i.i.d. case where $\rho$ is known, in the logistic noise model, we also consider estimating the noise level $\tau$ and evaluate the likelihood with respect to the noisy distribution. As $\tau$ increases, estimating the noise becomes more challenging, leading to higher sample complexity.

Finally, despite the problem's non-convexity, we demonstrate a practical algorithm for learning weighted power mean functions across tasks using simulated data and a real-world food resource allocation dataset from Lee et al. [11]. Our empirical results validate theoretical bounds and highlight algorithm performance across parameter settings.

**Related work.** Conceptually, our work is related to that of Procaccia et al. [18], who study the learnability of decision rules that aggregate individual utilities. In their work, however, individual utilities are represented as rankings over a set of alternatives (rather than cardinal utilities as in our case), and the rule to be learned is a voting rule that maps input rankings to a winning alternative. They provide sample complexity results for two families of voting rules: positional scoring rules and voting trees.

Basu and Echenique [1] derive VC dimension bounds for additive, Choquet, and max-min expected utility for decision-making under uncertainty, bounding the number of pairwise comparisons needed to falsify a candidate decision rule and estabiling learnability for these classes. Their work addresses decision rules operating on probability distributions rather than utility vectors, resulting in technical distinctions from ours; for instance, the max-min rule is not learnable in their setting (infinite VC dimension), whereas it is learnable in ours.

Kalai [10] studies the learnability of choice functions and establishes PAC guaranteees. Choice functions are defined with respect to a fixed and finite set of alternatives $X$, with each sample being a subset from $X$ and the choice over this subset. In contrast, our approach involves learning a function over an infinite action space where utilities are known.

Pellegrini et al. [17] conducts experiments on learning aggregation functions which are assumed to be a composition of $L_p$ means, observing that they perform favorably in various tasks such as scalar aggregation, set expansion and graph tasks. Our work provides a more theoretical analysis, proving the sample-efficient learnability of weighted power mean aggregation functions.

Melnikov and Hüllermeier [14] considers learning from actions with feature vectors and their global scores, with local scores for each individual unavailable for learning. They learn both local and global score functions, and consider the ordered weighted averaging operator for aggregating local scores. While we assume that each individual's local score is given, the aggregation function belongs to a richer function family motivated by social choice theory.

## 2   Problem Setup

We assume that the decision-making process concerns $d$ individuals. The decision-making setting we consider has each action associated with a positive utility vector $\mathbf{u} \in [u_{\min}, u_{\max}]^d \subset \mathbb{R}_+^d$, which describes the utilities derived from the $d$ individuals.

We encode the impact of each individual $i \in [d]$ on the decision-making process through a weight value $w_i \geq 0$ such that $\sum_{i=1}^d w_i = 1$. These weight values together form a weight vector $\mathbf{w} \in \Delta_{d-1}$. The weight vector might be a known or unknown quantity. A common instance in which the weight vector is known is when all agents are assumed to have an equal say, in which case $\mathbf{w} = \mathbf{1}_d/d$. For all settings we consider, we provide PAC guarantees for both known weights and unknown weights.

We assume that the decision-making process provides a cardinal social welfare value to each action. However, this social welfare value can be latent and need not be available to us as data. For the first task concerned with cardinal decision values, the social welfare values are available and can be used for learning. For the second task, both actions in the pair have a latent social welfare which is not available to us; however, the preferred action in the pair is known to us. We consider learning bounds with the empirical risk minimization (ERM) algorithm for all the losses in this work, with $\hat{p}$ being learned when the weights are known, and $(\hat{\mathbf{w}}, \hat{p})$ being learned when the weights are unknown.

**Power Mean.** The (weighted) power mean is defined on $p \in \mathbb{R} \cup \{\pm\infty\}$, and for $\mathbf{u} \in \mathbb{R}_+^d$, $\mathbf{w} \in \Delta_{d-1}$, it is

$$
M(\mathbf{u}; \mathbf{w}, p) = \begin{cases} \left( \sum_{i=1}^d w_i u_i^p \right)^{1/p} & p \neq 0 \\ \prod_{i=1}^d u_i^{w_i} & p = 0 \end{cases}
$$

It is sometimes more convenient to use the (natural) *log* power mean than the power mean. Since $\sum_{i=1}^{d} w_i = 1$, in effect we have $d$ variables, $w_1, \ldots, w_{d-1}$ and $p$. We refer to the weighted power mean family with known weight $\mathbf{w}$ as

$$\mathcal{M}_{\mathbf{w},d} = \{M(\cdot; \mathbf{w}, p) | p \in \mathbb{R}\}.$$

If the weight is unknown, the weighted power mean family is denoted by

$$\mathcal{M}_d = \{M(\cdot; \mathbf{w}, p) | p \in \mathbb{R}, \mathbf{w} \in \Delta_{d-1}\}.$$

The power mean family is a natural representation for social welfare functions. Cousins [7, 6] puts forward a set of axioms under which the set of possible welfare functions is precisely the weighted power mean family. An unweighted version of these functions results in the family of constant elasticity of substitution (CES) welfare functions [8], which are widely studied in econometrics.

To show the generality of this family of functions, we list a few illustrative cases:

- $M(\mathbf{u}; \mathbf{w}, p = -\infty) = \min_{i \in d} u_i$, which corresponds to egalitarian social welfare.

- $M(\mathbf{u}; \mathbf{w}, p = 0) = \prod_{i=1}^{d} u_i^{w_i}$, which corresponds to a weighted version of Nash social welfare.

- $M(\mathbf{u}; \mathbf{w}, p = 1) = \sum_{i=1} w_i u_i$, which corresponds to weighted utilitarian welfare.

- $M(\mathbf{u}; \mathbf{w}, p = \infty) = \max_{i \in d} u_i$, which corresponds to egalitarian social *welfare*.

We note that for $p = \pm\infty$, the decision utility is independent of $\mathbf{w}$. With $w_i = 1/d$ for all $i \in [d]$, we get the conventional interpretations of the welfare notions mentioned above.

The power mean family has some useful properties. An obvious one is that $M(\mathbf{u}, \mathbf{w}, p) \in [u_{(1)}, u_{(d)}]$, where $u_{(1)}$ and $u_{(d)}$ denote the first and $d$-th order statistics of $\mathbf{u} = (u_1, \ldots, u_n)$. $u_{(1)}$ is attained at $p = -\infty$, and $u_{(d)}$ is attained at $p = \infty$. A more general observation is the following:

**Lemma 2.1.** *(a) $M(\mathbf{u}; \mathbf{w}, p)$ is nondecreasing with respect to $p$ for all $\mathbf{u} \in [u_{\min}, u_{\max}]^d$, $\mathbf{w} \in \Delta_{d-1}$.*

*(b) $M(\mathbf{u}; \mathbf{w}, p)$ is monotonic with respect to $w_i$ for each $i \in [d-1]$, for all $\mathbf{u} \in [u_{\min}, u_{\max}]^d$, $p \in \mathbb{R}$.*

*(c) $M(\mathbf{u}; \mathbf{w}, p)$ and $\log M(\mathbf{u}; \mathbf{w}, p) - \log M(\mathbf{v}; \mathbf{w}, p)$ are quasilinear with respect to $\mathbf{w}$ if $p$ is fixed.*

This monotonicity of the power mean in $\mathbf{w}$ and $p$ was also noted by Qi et al. [19]. A proof for the above lemma is provided in Appendix A.1.

## 3 Cardinal Social Welfare

We first consider the case where we know the cardinal value of the social choice associated with each action. Learning in this setting thus corresponds to regression. Formally, we assume an underlying distribution $\mathcal{D} : [u_{\min}, u_{\max}]^d \times [u_{\min}, u_{\max}]$ over the utilities and social welfare values. We receive i.i.d samples $\{(\mathbf{u}_i, y_i)\}_{i=1}^{n} \sim \mathcal{D}^n$, $\mathbf{u}_i$ being the utility vector and $y_i \in [u_{i(1)}, u_{i(d)}]$ being the social welfare value associated with action $i$.

We consider the $\ell_2$ loss over $M(\mathbf{u}_i; \mathbf{w}, p)$ and $y_i$. The true risk in this case is

$$R(\mathbf{w}, p) = \mathbb{E}_{(\mathbf{u}, y) \sim \mathcal{D}} \left[ (M(\mathbf{u}; \mathbf{w}, p) - y)^2 \right].$$

To analyze the PAC learnability of this setting, we first provide bounds on the pseudo-dimensions[1] of $\mathcal{M}_{\mathbf{w},d}$ and $\mathcal{M}_d$. We begin by noting that

$$M(\mathbf{u}; \mathbf{w}, p) = u_{(d)} \cdot M(\mathbf{r}; \mathbf{w}, p), \quad \text{where } \mathbf{r} \in [d] \text{ and } r_i = \frac{u_i}{u_{(d)}}.$$

---

[1]For a formal definition of pseudo-dimension, refer to Definition A.2. A comprehensive review can be found in Vidyasagar and Vidyasagar [22].

Since $M(\mathbf{r}; \mathbf{w}, p) \in [0, 1]$, we can determine the pseudo-dimensions of this function class.

We now define the function classes

$$\mathcal{S}_{\mathbf{w},d} = \{f(\mathbf{u}; \mathbf{w}, p) = M(\mathbf{r}; \mathbf{w}, p) \mid (\mathbf{w}, p) \in \Delta_{d-1} \times \mathbb{R}\},$$
$$\mathcal{S}_d = \{f(\mathbf{u}; \mathbf{w}, p) = M(\mathbf{r}; \mathbf{w}, p) \mid p \in \mathbb{R}\}.$$

We then have the following bounds on pseudo-dimensions:

**Lemma 3.1.** *(a) If $\mathbf{w}$ is known, then $Pdim(\mathcal{S}_{\mathbf{w},d}) = 1$.*

*(b) If $\mathbf{w}$ is not known, then $Pdim(\mathcal{S}_d) < 8d(\log_2 d + 1)$.*

A detailed proof is provided in Appendix A.3.

We highlight the fact that $p$ and $\mathbf{w}$ are the parameters of the log power mean function family, which calls for the novel bounds provided in this work. These bounds on the pseudo-dimensions can now be used to obtain PAC bounds:

**Theorem 3.2.** *Given a set of samples $\{(\mathbf{u}_i, y_i)\}_{i=1}^n$ drawn from a distribution $\mathcal{D}^n$, for any $\delta > 0$, the following holds with probability at least $1 - \delta$ with respect to the $\ell_2$ loss function:*
*(a) If $\mathbf{w}$ is known, then*

$$R(\mathbf{w}, \hat{p}) - \inf_{p \in \mathbb{R}} R(\mathbf{w}, p) \leq 16\xi \left( \sqrt{\frac{2 \log 2 + 2 \log n}{n}} + \frac{c}{\sqrt{n}} \right) + 6\sqrt{\frac{\log(4/\delta)}{2n}}$$

*(b) If $\mathbf{w}$ is unknown, then*

$$R(\hat{\mathbf{w}}, \hat{p}) - \inf_{(\mathbf{w},p) \in \Delta_{d-1} \times \mathbb{R}} R(\mathbf{w}, p) \leq 16\xi \left( \sqrt{\frac{2 \log 2 + 16(d \log_2 d + 1) \log n}{n}} + \frac{c}{\sqrt{n}} \right)$$
$$+ 6\sqrt{\frac{\log(4/\delta)}{2n}}$$

*where $\xi = u_{\max} (u_{\max} - u_{\min})$.*
For the complete proof, see Appendix A.5. Below, we provide a proof sketch.

*Proof Sketch.* We first use the pseudo-dimensions found above to bound the Rademacher complexity of $\mathcal{M}_d$ and $\mathcal{M}_{\mathbf{w},d}$ in Lemma A.6. Since $M(\mathbf{u}_i; \mathbf{w}, p) \in [u_{\min}, u_{\max}]$ and $y_i \in [u_{\min}, u_{\max}]$, the $\ell_2$ loss function in this case has domain $[u_{\min} - u_{\max}, u_{\max} - u_{\min}]$. It is Lipschitz continuous on this domain with Lipschitz constant $2\xi$. Using Lemma A.6 and Talagrand's contraction lemma, we obtain the bounds

$$\hat{\mathfrak{R}}(\ell \circ \mathcal{M}_{\mathbf{w},d}) \leq 2(u_{\max} - u_{\min})\hat{\mathfrak{R}}(\mathcal{M}_{\mathbf{w},d}) \quad \text{and} \quad \hat{\mathfrak{R}}(\ell \circ \mathcal{M}_d) \leq 2(u_{\max} - u_{\min})\hat{\mathfrak{R}}(\mathcal{M}_d).$$

These Rademacher complexity bounds are then used to obtain the uniform convergence bounds above. □

These bounds are distribution-free, with the only assumption being that all utilities and social welfare values are in the range $[u_{\min}, u_{\max}]$. They also imply an $\mathcal{O}(1)$ and $\mathcal{O}(d \log d)$ dependence of sample complexity on $d$ for known and unknown weights respectively. Moreover, we observe the dependence of the upper bound on $u_{\max} - u_{\min}$ for the $\ell_2$ loss. We note that when $u_{\max} = u_{\min} = u_0$, all utilities and social welfare function values are also $u_0$. In this case, the Rademacher complexity bound is also zero, which is expected.

Computationally, $M(\mathbf{u}; \mathbf{w}, p)$ is non-convex in $\mathbf{w}$ and $p$, which means that the $\ell_2$ loss is also non-convex. However, we observe that from Lemma 2.1 (c), $M(\mathbf{u}; \mathbf{w}, p)$ is quasilinear w.r.t. $\mathbf{w}$ with fixed $p$, which makes the $\ell_2$ loss function quasi-convex for all $(\mathbf{u}, y)^2$. We use this fact to construct a practical algorithm.

A shortcoming of this setting is that decision-makers are required to provide a social welfare value for each action. A more natural setting might be when decision-makers only provide their preferences between actions — potentially just their *revealed* preferences, i.e., the choices they have made in the past — and we address this case next.

---

[2]A detailed explanation of the quasi-convexity of $\ell_2$ loss is provided in Appendix A.2.1.

# 4 Pairwise Preference Between Actions

For this setting, we assume an underlying distribution $\mathcal{D} : [u_{\min}, u_{\max}]^d \times [u_{\min}, u_{\max}]^d \times \{\pm 1\}$. We obtain i.i.d. samples $\{((\mathbf{u}_i, \mathbf{v}_i), y_i)\}_{i=1}^n \sim \mathcal{D}^n$, where $(\mathbf{u}_i, \mathbf{v}_i)$ are the utilities for the $i$-th pair of actions, and $y_i$ is a comparison between their (latent) social choice values. We encode the comparison function as $C : [u_{\min}, u_{\max}]^d \times [u_{\min}, u_{\max}]^d \to \{\pm 1\}$, with

$$C((\mathbf{u}, \mathbf{v}); \mathbf{w}, p) = \text{sign}\left(\log M(\mathbf{u}; \mathbf{w}, p) - \log M(\mathbf{v}; \mathbf{w}, p)\right).$$

We denote the family of above functions by $\mathcal{C}_{\mathbf{w},d} = \{C((\mathbf{u}, \mathbf{v}); \mathbf{w}, p) : p \in \mathbb{R}\}$ when the weights are known, and $\mathcal{C}_d = \{C((\mathbf{u}, \mathbf{v}); \mathbf{w}, p) : p \in \mathbb{R}, \mathbf{w} \in \Delta_{d-1}\}$ when the weights are unknown. We consider learning with $0 - 1$ loss over $C((\mathbf{u}_i, \mathbf{v}_i); \mathbf{w}, p)$ and $y_i$. The true risk in this case is

$$R(\mathbf{w}, p) = \mathbb{E}_{((\mathbf{u}, \mathbf{v}), y) \sim \mathcal{D}}\left[\frac{(1 + y \cdot C((\mathbf{u}, \mathbf{v}); \mathbf{w}, p))}{2}\right].$$

To provide convergence guarantees for the above setting, we bound the VC dimension of the comparison-based function classes mentioned above

**Lemma 4.1.** *(a) If $\mathbf{w}$ is known, then $VC(\mathcal{C}_{\mathbf{w},d}) < 2(\log_2 d + 1)$.*
*(b) If $\mathbf{w}$ is unknown, then $VC(\mathcal{C}_d) < 8(d \log_2 d + 1)$.*
*(c) (Lower bounds): $VC(\mathcal{C}_d) \geq \log_2 d + 1$, and $VC(\mathcal{C}_{\mathbf{w},d}) \geq d - 1$*

The detailed proof of the above lemma is provided in Appendix A.6.

We find the asymptotically tight lower bound for the known weights case rather surprising, as it is *a priori* unclear that the correct bound should be superconstant and scale with $d$.

The finiteness of VC dimension guarantees PAC learnability, and we get uniform convergence bounds using the VC theorem.

**Theorem 4.2.** *Given samples $\{((\mathbf{u}_i, \mathbf{v}_i), y_i)\}_{i=1}^n \sim \mathcal{D}^n$ and for 0-1 loss and any $\delta > 0$, with probability at least $1 - \delta$,*
*(a) If $\mathbf{w}$ is known, then*

$$R(\mathbf{w}, \hat{p}) - \inf_{p \in \mathbb{R}} R(\mathbf{w}, p) \leq 16\sqrt{\frac{2(\log_2 d + 1)\log(n+1) + \log(8/\delta)}{n}}$$

*(b) If $\mathbf{w}$ is unknown, then*

$$R(\hat{\mathbf{w}}, \hat{p}) - \inf_{(\mathbf{w}, p) \in \Delta_{d-1} \times \mathbb{R}} R(\mathbf{w}, p) \leq 16\sqrt{\frac{8(d \log_2 d + 1)\log(n+1) + \log(8/\delta)}{n}}$$

We note that unlike the bounds on $\ell_2$ loss of Theorem 3.2, these bounds on 0-1 loss are independent of the range of utility values and only depend on $d$. They provide sample complexity bounds which depend on $d$ as $\mathcal{O}(\log d)$ and $\mathcal{O}(d \log d)$ for known and unknown weights respectively. Despite these PAC guarantees, empirical risk minimization can be particularly difficult in this case, since the loss function as well as the function class $\log M(\mathbf{u}; \mathbf{w}, p) - \log M(\mathbf{v}; \mathbf{w}, p)$ can be non-convex. To illustrate this non-convexity, we plot the value of the above function for two pairs of utility vectors with respect to $p$ in Figure 6, with $d = 6$ and $\mathbf{w} = \mathbf{1}_d/d$. However, the quasilinearity of $\log M(\mathbf{u}; \mathbf{w}, p) - \log M(\mathbf{v}; \mathbf{w}, p)$ with fixed $p$ can be used to design efficient algorithms.

## 4.1 Convergence Bounds Under I.I.D Noise

Decision making can be especially challenging if two actions are difficult to compare, and the preference data we obtain can potentially be noisy. We first consider each comparison to be mislabeled in an i.i.d. manner with known probability $\rho \in [0, 1/2)$. We make use of the framework developed by Natarajan et al. [16], and we consider convergence guarantees under 0-1 loss.

Specifically, the unbiased estimator of $\ell_{0-1}$ is

$$\tilde{\ell}_{0-1}(t, y) = \frac{(1-\rho)\ell_{0-1}(t, y) - \rho\ell_{0-1}(t, -y)}{1 - 2\rho}.$$

We conduct ERM with respect to $\tilde{\ell}_{0-1}$ to obtain $(\hat{\mathbf{w}}, \hat{p}) \in \Delta_{d-1} \times \mathbb{R}$ (only learning $p$ if weights are known). We observe that $\ell_{0-1}(t, y) = (1 + ty)/2$ is 1/2-Lipschitz in $t$, $\forall t, y \in \{\pm 1\}$. Using Theorem 3 of Natarajan et al. [16], we get the following convergence bounds:

**Theorem 4.3.** *Given samples* $\{((\mathbf{u}_i, \mathbf{v}_i), y_i)\}_{i=1}^n \sim \mathcal{D}^n$, *for any* $\delta > 0$ *and for any* $\rho \in [0, 1/2)$, *with probability at least* $1 - \delta$ *with respect to 0-1 loss,*

   *(a) If* $\mathbf{w}$ *is known, then*

$$R(\hat{\mathbf{w}}, \hat{p}) - \inf_{p \in \mathbb{R}} R(\mathbf{w}, p) \leq \frac{8}{1 - 2\rho} \sqrt{\frac{(\log_2 d + 1) \log(n + 1)}{n}} + 2\sqrt{\frac{\log(1/\delta)}{2n}}$$

   *(b) If* $\mathbf{w}$ *is unknown, then*

$$R(\hat{\mathbf{w}}, \hat{p}) - \inf_{(\mathbf{w}, p) \in \Delta_{d-1} \times \mathbb{R}} R(\mathbf{w}, p) \leq \frac{16}{1 - 2\rho} \sqrt{\frac{(d \log_2 d + 1) \log(n + 1)}{n}} + 2\sqrt{\frac{\log(1/\delta)}{2n}}$$

A detailed proof of the above theorem is provided in Appendix A.7.

We note that although ERM is conducted with respect to $\tilde{\ell}_{0-1}$ on the noisy distribution, the risks are defined on the underlying noiseless distribution. This gives $\mathcal{O}(\log d/(1 - 2\rho)^2)$ and $\mathcal{O}(d \log d/(1 - 2\rho)^2)$ sample complexities for the known and unknown weights cases respectively. We note that when $\rho = 0$, the above bounds reduce to the noiseless bounds in Theorem 4.2. Since the noise level $\rho$ is usually not known to us, it can be estimated using cross-validation as suggested by Natarajan et al. [16].

However, conducting ERM on $\tilde{\ell}_{0-1}$ might be prohibitively difficult due to the non-convex nature of the function. An i.i.d noise model might also be inappropriate in certain settings; we next consider a more natural noise model.

## 5 Pairwise Preference With Logistic Noise

Intuitively, we expect that two actions would be harder to compare if their social welfare values are closer to each other. We formalize this intuition in the form of a noise model inspired by the BTL noise model [4, 13]. Let $\mathbf{w}^*$ and $p^*$ be the true power mean parameters, and let $\tau^* \in [0, \tau_{\max}]$ be a temperature parameter. For an action pair $(\mathbf{u}, \mathbf{v})$, we assume that the probability of $\mathbf{u}$ being preferred to $\mathbf{v}$ is

$$\mathbb{P}(y = 1 | (\mathbf{u}, \mathbf{v}); \mathbf{w}^*, p^*, \tau^*) = \frac{1}{1 + \exp\left(-\tau^* (\log M(\mathbf{u}; \mathbf{w}^*, p^*) - \log M(\mathbf{v}; \mathbf{w}^*, p^*))\right)} \quad (1)$$

We see that a larger difference between the log power means of $\mathbf{u}$ and $\mathbf{v}$ translates to a higher probability of $\mathbf{u}$ being preferred. If $\mathbf{u}$ and $\mathbf{v}$ lie on the same level set of $\log M(\cdot; \mathbf{w}^*, p^*)$, the probability becomes 0.5, which matches the intuition of both actions being equally preferred. We also note the dependence of the probability on $\tau^*$: a higher $\tau^*$ corresponds to more confidence in the preferences, with $\tau^* = 0$ meaning indifference for all pairs of actions. The mislabeling probability is also invariant to scaling of $\mathbf{u}$ and $\mathbf{v}$.

Our learning task now becomes estimating $\mathbf{w}$, $p$ and $\tau$ given data. We denote the function family in this case by

$$\mathcal{T}_{\mathbf{w},d} = \{\tau (\log M(\cdot; \mathbf{w}, p) - \log M(\cdot; \mathbf{w}, p)) \,|\, \tau, p\}$$

when the weights are known, and

$$\mathcal{T}_d = \{\tau (\log M(\cdot; \mathbf{w}, p) - \log M(\cdot; \mathbf{w}, p)) \,|\, \tau, \mathbf{w}, p\}$$

when the weights are unknown. A natural loss function to consider in this case is negative log likelihood, and we consider PAC learnability with this loss. Using the framework developed in Section 3, we obtain the following PAC bounds:

**Theorem 5.1.** *Given samples* $\{((\mathbf{u}_i, \mathbf{v}_i), y_i)\}_{i=1}^n \sim \mathcal{D}^n$ *and for negative log likelihood loss, for all* $\delta > 0$, *with probability at least* $1 - \delta$,
   *(a) If* $\mathbf{w}$ *is known, then*

$$R(\mathbf{w}, \hat{p}) - \inf_{p \in \mathbb{R}} R(\mathbf{w}, p) \leq 16\tau_{\max}\kappa \left(\sqrt{\frac{2 \log 2 + 2 \log n}{n}} + \frac{c}{\sqrt{n}}\right) + 6\sqrt{\frac{\log(4/\delta)}{2n}}$$

*(b) If $\mathbf{w}$ is unknown, then*

$$R(\hat{\mathbf{w}}, \hat{p}) - \inf_{(\mathbf{w},p) \in \Delta_{d-1} \times \mathbb{R}} R(\mathbf{w}, p) \leq 16\tau_{\max}\kappa\sqrt{\frac{2\log 2 + 16(d\log_2 d + 1)\log n}{n}}$$

$$+ 16\tau_{\max}\kappa\frac{2c}{\sqrt{n}} + 3\sqrt{\frac{\log(4/\delta)}{2n}}$$

*where* $\kappa = \log(u_{\max}/u_{\min})$.

We derive this result in detail in Appendix A.8.

This gives us sample complexity bounds of $\mathcal{O}(1)$ and $\mathcal{O}(d\log d)$ with respect to $d$ for the known and unknown weights cases respectively, thus establishing PAC learnability. An important distinction between Theorem 4.3 and the above theorem is that Theorem 4.3 bounds risk with respect to 0-1 loss, while the above theorem bounds risk with respect to logistic loss which is continuous and hence easier to control. Moreover, we estimate the noise level $\tau$ in the logistic case along with $\mathbf{w}$ and $p$, whereas Theorem 4.3 is concerned with estimating $\mathbf{w}$ and $p$.

As with the previous cases, non-convexity in this setting also makes global optimization with respect to $\mathbf{w}$ and $p$ (and hence ERM) difficult. We observe that logistic loss is quasilinear in $\mathbf{w}$ with fixed $p$[3], and this observation can be used to construct an effective algorithm.

## 6   Empirical Results

We conduct several simulations on semi-synthetic data to gain additional insight into sample complexity and demonstrate an empirically effective algorithm. The implementation also serves to demonstrate the practicability of our approach, including the availability of individual utility functions.

**Data.** The dataset we rely on (which is not publicly available) comes from the work of Lee et al. [11] with a US-based nonprofit that operates an on-demand donation transportation service supported by volunteers. WeBuildAI is a participatory framework that enables stakeholders, including donors, volunteers, recipient organizations, and nonprofit staff, to collaboratively design algorithmic policies for allocating donations. Donors provide food donations, volunteers transport the donations, recipient organizations receive and distribute the food, and dispatchers (nonprofit staff) manage the allocation and logistics. The "actions" are hundreds of recipient organizations that may receive an incoming donation.

As part of this framework, Lee et al. [11] learned a (verifiably realistic) utility function over the actions for each of 18 stakeholders from the different groups based on 8 features: travel time between donors and recipients, recipient organization size, USDA-defined food access levels in recipient neighborhoods, median household income, poverty rates, the number of weeks since the last donation, the total number of donations received in the last three months, and the type of donation (common or uncommon).

In our simulations, we use the values of these stakeholder utility functions learned by Lee et al. [11] as the utility vectors. We fix a $p^*$ and weight vector $\mathbf{w}^*$ to generate the social welfare values $M(\mathbf{u}; \mathbf{w}, p)$. We use noisy versions of these social welfare values in the cardinal case, whereas noisy pairwise comparisons between random pairs of utility vectors are used in the ordinal case.

**Algorithm.** As noted in previous sections, $\ell_2$ and logistic losses are quasiconvex with respect to $\mathbf{w}$ for single samples when $p$ is fixed. Although the sum of quasiconvex functions is not guaranteed to be quasiconvex, we empirically observe that gradient descent on the loss function applied to the data can still lead to convergence to a minimum which has empirical risk comparable to that of the true parameters. As our simulations show, this minimum increasingly resembles $\mathbf{w}^*$ (the real weight) with decreasing noise. Thus, our algorithm consists of performing a grid search on $p$ and conducting gradient descent on $\mathbf{w}$ for each $p$. We provide more details about the algorithm in Appendix B.

**Cardinal case.** We consider $p^* = 2.72$ and a random weight $\mathbf{w}^*$. We then add Gaussian noise with standard deviation $\left(u_{i(d)} - u_{i(1)}\right) \cdot \nu$ to each sample, where $\nu$ corresponds to the noise level. The Gaussian noise is clipped to stay within $\left[u_{i(1)}, u_{i(d)}\right]$. Finally, we learn $p$ and $\mathbf{w}$ using our algorithm, and we present the results in Figure 1.

---

[3]A detailed explanation of this fact is provided in Appendix A.2.2

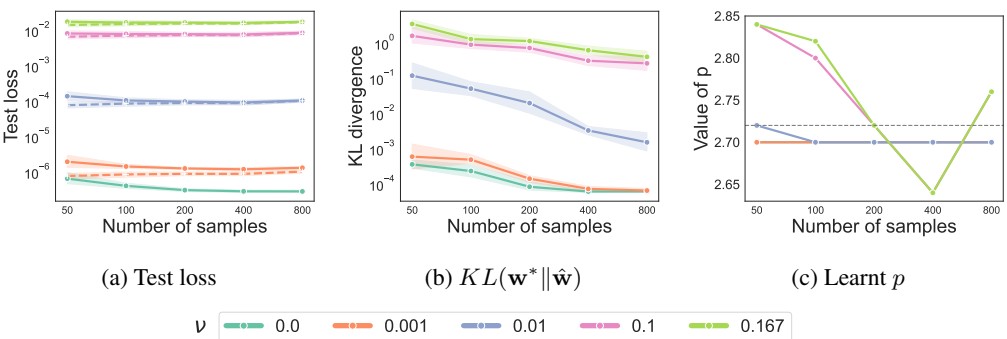

(a) Test loss         (b) $KL(\mathbf{w}^*\|\hat{\mathbf{w}})$         (c) Learnt $p$

Figure 1: Results for cardinal case with number of samples. Different lines show results for different values of added noise $\nu$. Solid lines correspond to values for learned parameters, whereas dotted lines correspond to values for real parameters.

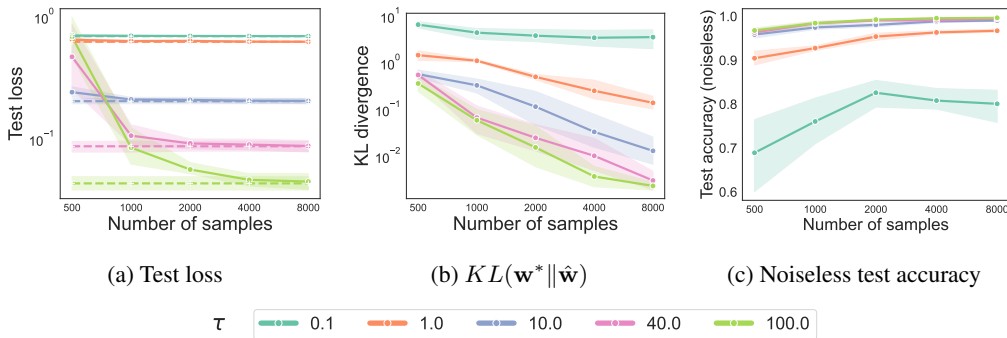

(a) Test loss         (b) $KL(\mathbf{w}^*\|\hat{\mathbf{w}})$         (c) Noiseless test accuracy

Figure 2: Results for ordinal case with number of samples. Different lines show results for different values of noise level $\tau$. Solid lines correspond to values for learned parameters, whereas dotted lines correspond to values for real parameters.

In Figure 1a, we observe that the test loss for learned parameters decreases with decreasing noise and increasing number of samples. We also observe that the test loss for learned parameters closely matches that for real parameters in Figure 1a. In Figure 1b, we observe that KL divergence between the true and learnt weights decreases uniformly with decreasing noise and increasing number of samples. This supports the fact that our algorithm is indeed able to find the correct minimum. We also plot the trend of mean learned $p$ in Figure 1c, and we observe that the learned $p$ increasingly resembles the real $p^*$ with lower noise and greater number of samples. Plots for train loss and loss on noiseless test data are provided in Appendix C.

**Ordinal case.** We consider $p^* = -1.62$ and a random weight $\mathbf{w}^*$. We compare each sample in the considered training data with 10 other randomly chosen samples, with the comparisons being noised according to the logistic noise model in Equation (1). We then learn $\mathbf{w}$ and $\tau$ for each $p$ in the chosen grid and then choose the best $p$. Our results are shown in Figure 2.

In Figure 2a we observe that the test loss for learned parameters matches that for real parameters for small $\tau^*$ and a large number of training samples. The relative deviation between test losses progressively increases for smaller numbers of samples and smaller $\tau^*$. We note that small $\tau^*$ corresponds to more noise in the comparisons, which results in higher losses. However, the deviation between learned loss and true loss is smaller, as we are also estimating the noise parameter, which is easier to estimate for small $\tau^*$, since the logistic function has a larger gradient.

We observe a uniform decrease in KL divergence between $\mathbf{w}$ and $\mathbf{w}^*$ for a larger number of samples and smaller $\tau^*$, again pointing to the effectiveness of the algorithm. We also observe that test accuracy on noiseless data increases with more samples and higher $\tau^*$. Interestingly, for $\tau^* = 0.1$ and $\tau^* = 1$, the test accuracy on noiseless data (Figure 2c) is significantly higher than that on (noisy) test data, another indicator of effective ERM being conducted by the algorithm.

In $\href{}{\text{Figure 4b}}$ in $\href{}{\text{Appendix C}}$, we observe greater variation in learned $p$ compared to the cardinal case. A possible reason behind this is that changes in $p$ result in smaller changes in losses for negative $p$ than for positive $p$. This hypothesis is supported by simulations for the ordinal case conducted for $p = 1.62$, with results presented in Figure 5. In Figure 5e, we observe that learned $p$ is much more consistent with the real $p$ as $\tau^*$ decreases.

We also conduct simulations on fully synthetic data to study the effect of $d$, and we present the results in Appendix E. We verify the theoretical $\mathcal{O}(d \log d)$ scaling of error with unknown weights for the ordinal case in Figure 8.

## 7 Discussion

Our work has (at least) several limitations, which can inspire future work. First, as seen in Section 6, we are able to gain access to realistic utility vectors, in this case ones based on models that were learned from pairwise comparisons. Utilities are also routinely estimated for other economically-motivated algorithms — say, Stackelberg security games [21].

However, these estimates are of course not completely accurate. It is an interesting direction of future work to extend our results to the setting where the utility vectors need to be estimated, either by an outside expert, or using input from the individuals themselves.

Although our experiments demonstrate convergence of the algorithm to the correct minimum, rigorous theoretical analysis about the nature of minima for the $\ell_2$ and logistic loss functions is still needed and could lead to algorithmic improvements. One issue is that scaling the algorithm to the national scale – $d = 10^8$, say, can be prohibitively expensive.

Finally, our work only applies to weighted power mean functions. While we have argued that this family is both expressive and natural, it would be exciting to obtain results for even broader, potentially non-parametric families of social welfare functions.

The ability to learn social welfare functions can enable us to understand a decision maker's priorities and ideas of fairness, based on past decisions they have made. This has direct societal impact as these notions can be used to both understand biases and inform the design of improved fairness metrics. A second potential application is to imitate a successful decision maker's policy choices in future dilemmas or in other domains. This may pose some ethical questions if the learning model is misspecified; however, the restriction of the function class to weighted power means, which is inspired by natural social choice theory axioms, mitigates this risk.

## Acknowledgements

This research was partially supported by the National Science Foundation under grants IIS-2147187, IIS-2229881, and CCF-2007080; by the Office of Naval Research under grants N00014-20-1-2488, N00014-24-1-2704, and N00014-22-1-2181; and by the USDA under grant 2021-67021-35329.

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

# A Deferred Proofs

## A.1 Proof of Lemma 2.1

*Proof of Lemma 2.1 (a).* Let $p > 0$. Define $t_i = \frac{u_i}{u_{\max}}$ for all $i \in [n]$. Since $t_i \leq 1$ for all $i \in [n]$ and given that $\sum_{i=1}^{n} w_i = 1$, it follows that $\sum_{i=1}^{n} w_i t_i^p \leq \sum_{i=1}^{n} w_i = 1$. Therefore, $\log(w_i t_i^p) \leq 0$ for each $i$. Given $p > q > 0$, we obtain

$$
0 \leq \log \left( \sum_{i=1}^{d} w_i t_i^p \right) (p^{-1} - q^{-1})
$$

$$
= p^{-1} \log \left( \sum_{i=1}^{d} w_i t_i^p \right) - q^{-1} \log \left( \sum_{i=1}^{d} w_i t_i^p \right)
$$

$$
\leq p^{-1} \log \left( \sum_{i=1}^{d} w_i t_i^p \right) - q^{-1} \log \left( \sum_{i=1}^{d} w_i t_i^q \right)
$$

$$
= \left( u_{\max} + p^{-1} \log \sum_{i=1}^{d} w_i t_i^p \right) - \left( u_{\max} + q^{-1} \log \sum_{i=1}^{d} w_i t_i^q \right)
$$

$$
= \log M(\mathbf{u}; \mathbf{w}, p) - \log M(\mathbf{u}; \mathbf{w}, q).
$$

This derivation similarly holds for the case $0 > p > q$, demonstrating the monotonicity of $\log M(\mathbf{u}; \mathbf{w}, p)$ with respect to $p$. The continuity of $\log M(\mathbf{u}; \mathbf{w}, p)$ with respect to $p$ at $p = 0$ ensures the monotonicity for all $p \in \mathbb{R}$. Since $\log$ is a strictly increasing function, this guarantees the monotonicity of $M(\mathbf{u}; \mathbf{w}, p)$. $\square$

*Proof of Lemma 2.1 (b).* Since $\sum_{i=1}^{d} w_i = 1$, we express $w_d = 1 - \sum_{i=1}^{d-1} w_i$ and consider $d - 1$ variables.

For $p \neq 0$, we have

$$
\log M(\mathbf{u}; \mathbf{w}, p) = p^{-1} \log \left( \sum_{i=1}^{d-1} w_i u_i^p + u_d^p \left( 1 - \sum_{i=1}^{d-1} w_i \right) \right)
$$

$$
\implies \frac{\partial \log M(\mathbf{u}; \mathbf{w}, p)}{\partial w_i} = p^{-1} \left( \frac{u_i^p - u_d^p}{\sum_{i=1}^{d-1} w_i u_i^p + u_d^p \left( 1 - \sum_{i=1}^{d-1} w_i \right)} \right)
$$

$$
= \left( \frac{u_i^p - u_d^p}{\sum_{i=1}^{d} w_i u_i^p} \right) p^{-1}.
$$

$\sum_{i=1}^{d} w_i u_i^p$ is positive since it is a positive weighted sum of utilities. We aim to show that

$$
p^{-1}(u_i^p - u_d^p) > 0.
$$

Suppose that $u_i > u_d$. If $p > 0$, then $u_i^p - u_d^p > 0$. Conversely, if $p < 0$, then $u_i^p - u_d^p < 0$, but since $p^{-1}$ is also negative, the product remains positive. Thus, if $u_i > u_d$, the log-norm increases with $w_i$. A similar argument shows that the log-norm decreases if $u_i < u_d$.

For $p = 0$, we have

$$
\frac{\partial \log M(\mathbf{u}; \mathbf{w}, 0)}{\partial w_i} = \frac{\partial}{\partial w_i} \left( \sum_{i=1}^{d-1} w_i \log u_i + \left( 1 - \sum_{i=1}^{d-1} w_i \right) \log u_d \right)
$$

$$
= \log \left( \frac{u_i}{u_d} \right)
$$

$$
= \lim_{p \to 0} \frac{\partial \log M(\mathbf{u}; \mathbf{w}, p)}{\partial w_i}
$$

This indicates that for $u_i > u_d$, the derivative is positive, implying an increase, and negative for $u_i < u_d$, implying a decrease. Thus, the function is monotonic for all $w_i \in [d-1]$. Since $\log$ is a strictly increasing function, this guarantees the monotonicity of $M(\mathbf{u}; \mathbf{w}, p)$. $\qquad \square$

*Proof of Lemma 2.1 (c).* We prove that $\log M(\mathbf{u}; \mathbf{w}, p) - \log M(\mathbf{v}; \mathbf{w}, p)$ is quasilinear. The proof for $M(\mathbf{u}; \mathbf{w}, p)$ follows by setting $\mathbf{v} = \mathbf{1}_d$.

As noted in [3], the ratio of two linear functions is quasilinear when the denominator is strictly positive. Given that $\langle \mathbf{w}, \mathbf{v}^p \rangle > 0$ for all $\mathbf{w} \in \Delta_{d-1}$, it follows that

$$f(\mathbf{w}) = \frac{\langle \mathbf{w}, \mathbf{u}^p \rangle}{\langle \mathbf{w}, \mathbf{v}^p \rangle}$$

is a quasilinear function. Furthermore, since $f(\mathbf{w}) > 0$ for all $\mathbf{w} \in \Delta_{d-1}$, and for any $x > 0$, $x^{1/p}$ is monotone for $p \in \mathbb{R} \setminus \{0\}$, the expression $M(\mathbf{u}; \mathbf{w}, p)/M(\mathbf{v}; \mathbf{w}, p) = f(\mathbf{w})^{1/p}$ is also quasilinear. Because $g(x) = \log(x)$ is monotone, quasilinearity is preserved, which implies that $\log M(\mathbf{u}; \mathbf{w}, p) - \log M(\mathbf{v}; \mathbf{w}, p)$ is quasilinear as well. $\qquad \square$

## A.2 Properties of Loss Functions

### A.2.1 Quasiconvexity of $\ell_2$ Loss

Since $M(\mathbf{u}; \mathbf{w}, p)$ is quasilinear by Lemma 2.1 (c), it follows that $f(\mathbf{w}) = M(\mathbf{u}; \mathbf{w}, p) - y$ is also quasilinear. For $\mathbf{w}_1, \mathbf{w}_2 \in \Delta_{d-1}$, we therefore have

$$\min \{f(\mathbf{w}_1), f(\mathbf{w}_2)\} \leq f(\lambda \mathbf{w}_1 + (1 - \lambda)\mathbf{w}_2) \leq \max \{f(\mathbf{w}_1), f(\mathbf{w}_2)\},$$
$$\implies f(\lambda \mathbf{w}_1 + (1 - \lambda)\mathbf{w}_2)^2 \leq \max \{f(\mathbf{w}_1)^2, f(\mathbf{w}_2)^2\}.$$

Thus, $f(\mathbf{w})^2 = (M(\mathbf{u}; \mathbf{w}, p) - y)^2$ is quasiconvex.

### A.2.2 Quasilinearity of Logistic Loss

By Lemma 2.1 (c), we know that $\log M(\mathbf{u}; \mathbf{w}, p) - \log M(\mathbf{v}; \mathbf{w}, p)$ is quasilinear. We consider two cases for the logistic loss. For $y = 1$, since $-\log \sigma(x) = \log(1 + \exp(-x))$ is a monotonic function, it preserves quasilinearity. For $y = 0$, we have $-\log(1 - \sigma(x)) = \log(1 + \exp(-x)) + x$, which is also monotonic and therefore preserves quasilinearity.

Using these properties, we conclude that the logistic loss term $-y \log \sigma(x) - (1 - y) \log(1 - \sigma(x))$ is a quasilinear function.

## A.3 Proof of Lemma 3.1

We differentiate between two representations of Rademacher complexity:

$$\hat{\mathfrak{R}}(\mathcal{F}) = \frac{1}{n} \mathbb{E}_\epsilon \left[ \sup_{f \in \mathcal{F}} \sum_{i=1}^{n} \epsilon_i f(x_i) \right],$$

$$\hat{\mathfrak{R}}_{abs}(\mathcal{F}) = \frac{1}{n} \mathbb{E}_\epsilon \left[ \sup_{f \in \mathcal{F}} \left| \sum_{i=1}^{n} \epsilon_i f(x_i) \right| \right].$$

It is clear that $\hat{\mathfrak{R}}(\mathcal{F}) \leq \hat{\mathfrak{R}}_{abs}(\mathcal{F})$. We now list a few results related to Pollard's pseudo-dimension.

**Definition A.1** (Pseudo-shattering). Let $\mathcal{H}$ be a set of real valued functions from input space $\mathcal{X}$. We say $C = (x_1, \ldots, x_m)$ is pseudo-shattered if there exists a vector $r = (r_1, \ldots, r_m)$ such that for all $b \in \{\pm 1\}^m = (b_1, \ldots, b_m)$, there exists $h_b \in \mathcal{H}$ such that $\text{sign}(h_b(x_i) - r_i) = b_i$.

**Definition A.2** (Pseudo-dimension ). The pseudo-dimension $\text{Pdim}(\mathcal{H})$ is the cardinality of the largest set pseudo-shattered by $\mathcal{H}$.

The following lemma connects pseudo-dimensions to VC dimensions:

**Lemma A.3** (Mohri [15]). *For every $h \in \mathcal{H}$, define the binary function $B_h(x,r) = sign\,(h(x) - r)$. Define $B_\mathcal{H} = \{B_h : h \in \mathcal{H}\}$. Then, $VC(B_\mathcal{H}) = Pdim(\mathcal{H})$*

The following lemma bounds the Rademacher complexity using pseudo-dimension and covering numbers.

**Lemma A.4** (Mohri [15]). *For $\mathcal{F} \subseteq [0,1]^\mathcal{X}$ with $Pdim(\mathcal{F}) \leq d$,*

$$\mathcal{N}(\epsilon, \mathcal{F}, d_n) \leq \left(\frac{c}{\epsilon}\right)^{2d},$$

*where $d_n(f,g) = \left(\frac{1}{n}\sum_{i=1}^n (f(x_i) - g(x_i))^2\right)^{1/2}$.*

We also have the following covering number bound for Rademacher complexity:

**Lemma A.5** (Mohri [15]). *For $\mathcal{F} \subseteq [0,1]^\mathcal{X}$,*

$$\hat{\mathfrak{R}}_{abs}(\mathcal{F}) \leq \inf_{\epsilon > 0}\left(\sqrt{\frac{2\log 2\mathcal{N}(\epsilon, \mathcal{F}, d_n)}{n}} + \epsilon\right).$$

We now turn to bounding the complexity for the unknown and known weights cases:

*Proof of Lemma 3.1 (a).* The function class is:

$$\mathcal{M}_{\mathbf{w},d} = \{M(\mathbf{r}; \mathbf{w}, p)|p \in \mathbb{R}\}.$$

Moreover, from Lemma 2.1, it follows that for a fixed $\mathbf{u} \in \mathbb{R}^d$, $M(\mathbf{u}; \mathbf{w}, p)$ is a non-decreasing function with respect to $p$. Consequently, there exists a $p^* \in \mathbb{R} \cup \{\pm\infty\}$ such that for any $y \in (u_{\min}, u_{\max})$, we have $M(\mathbf{u}; \mathbf{w}, p) < y$ for all $p < p^*$, and $M(\mathbf{u}; \mathbf{w}, p) \geq y$ for all $p \geq p^*$. This implies that $B_M(\mathbf{u}, y) = \text{sign}(M(\mathbf{u}; \mathbf{w}, p) - y)$ changes its sign exactly once as $p$ increases.

We note that for $B_M(x, y)$, one point can be shattered (by choosing $p < p^*$ and $p > p^*$). However, for two points $\mathbf{u}$ and $\mathbf{v}$, the number of times a sign change occurs with increasing $p$ for either $\mathbf{u}$ or $\mathbf{v}$ is at most twice, meaning that only 3 labels can be achieved. Thus, 2 points cannot be shattered. $\square$

*Proof of Lemma 3.1 (b).* The function class is:

$$\mathcal{M}_d = \{M(\cdot; \mathbf{w}, p)|p \in \mathbb{R}\}.$$

We note that

$$\begin{aligned}
B_M(\mathbf{u}, y) &= \text{sign}\,(M(\mathbf{u}; \mathbf{w}, p) - y) \\
&= \text{sign}\,(\log M(\mathbf{u}; \mathbf{w}, p) - \log y) \\
&= \text{sign}\,(\log M(\mathbf{u}; \mathbf{w}, p) - \log M(y \cdot \mathbf{1}_d; \mathbf{w}, p))
\end{aligned}$$

which, we observe, is exactly the expression in the noiseless comparison-based setup for the unknown weights case. We show in Lemma 4.1 (b) that the VC dimension for this expression is upper bounded by $8(d\log_2 d + 1)$. Thus, our result is proved. $\square$

### A.4 Rademacher Complexity Bound for the Cardinal Case

**Lemma A.6.**   *(a) If $\mathbf{w}$ is known, then*

$$\hat{\mathfrak{R}}(\mathcal{M}_{\mathbf{w},d}) \leq u_{\max}\left(\sqrt{\frac{2\log 2 + 2\log n}{n}} + \frac{c}{\sqrt{n}}\right).$$

*(b) If $\mathbf{w}$ is unknown, then*

$$\hat{\mathfrak{R}}(\mathcal{M}_{\mathbf{w},d}) \leq u_{\max}\left(\sqrt{\frac{2\log 2 + 16(2\log_2 d + 1)\log n}{n}} + \frac{c}{\sqrt{n}}\right),$$

*where $c > 0$ is a constant.*

*Proof.* We prove the result for the case of unknown weights; the result for known weights follows by replacing the pseudo-dimension bound of $\mathcal{S}_d$ with that of $\mathcal{S}_{\mathbf{w},d}$ from Lemma 3.1. Let $d_p$ denote the pseudo-dimension in the unknown weights case.

$$
\begin{aligned}
\hat{\mathfrak{R}}(\mathcal{M}_d) &= \mathbb{E}_\epsilon \left[ \frac{1}{n} \sup_{(\mathbf{w},p)\in\Delta_{d-1}\times\mathbb{R}} \sum_{i=1}^n \epsilon_i M(\mathbf{u}_i;\mathbf{w},p) \right] \\
&= \mathbb{E}_\epsilon \left[ \frac{1}{n} \sup_{(\mathbf{w},p)\in\Delta_{d-1}\times\mathbb{R}} \sum_{i=1}^n \epsilon_i u_{\max} \cdot M(\mathbf{r}_i;\mathbf{w},p) \right] \\
&= u_{\max} \mathbb{E}_\epsilon \left[ \frac{1}{n} \sup_{(\mathbf{w},p)\in\Delta_{d-1}\times\mathbb{R}} \sum_{i=1}^n \epsilon_i M(\mathbf{r}_i;\mathbf{w},p) \right] \\
&\leq u_{\max} \mathbb{E}_\epsilon \left[ \frac{1}{n} \sup_{(\mathbf{w},p)\in\Delta_{d-1}\times\mathbb{R}} \sum_{i=1}^n |\epsilon_i M(\mathbf{r}_i;\mathbf{w},p)| \right] \\
&= u_{\max} \hat{\mathfrak{R}}_{\text{abs}}(\mathcal{S}_d).
\end{aligned}
$$

From Lemma A.4 and Lemma A.5, and given that $\log M(\mathbf{r};\mathbf{w},p) \in [0,1]$, we have

$$
\mathcal{N}(\epsilon, \mathcal{S}_d, d_n) \leq \left(\frac{c}{\epsilon}\right)^2 d_p,
$$

$$
\hat{\mathfrak{R}}_{\text{abs}}(\mathcal{S}_d) \leq \inf_{\epsilon>0}\left( \sqrt{\frac{2\log 2 + 4d_p\log(c/\epsilon)}{n}} + \epsilon \right)
$$

$$
\leq \sqrt{\frac{2\log 2 + 2d_p\log n}{n}} + \frac{c}{\sqrt{n}}. \qquad (\text{setting } \epsilon = c/\sqrt{n})
$$

thus giving

$$
\hat{\mathfrak{R}}(\mathcal{M}_d) = u_{\max}\left( \sqrt{\frac{2\log 2 + 2d_p\log n}{n}} + \frac{c}{\sqrt{n}} \right).
$$

Substituting $d_p = 8(d\log_2 d + 1)$ yields the required bound. $\qquad\square$

We observe that the above lemma provides bounds of $\mathcal{O}(\sqrt{\log(n)/n})$ for known weights and $\mathcal{O}(\sqrt{d\log(d)\log(n)/n})$ for unknown weights on the Rademacher complexity. Notably, these bounds depend on $u_{\max}$, indicating that the complexity of the function class grows as the maximum utility value increases.

### A.5   Proof for Theorem 3.2

*Proof.* We prove the result for the case of unknown weights; the result for known weights follows a similar argument. For the $\ell_2$ loss, the function class is defined as

$$
\mathcal{L}_2 = \left\{ \ell_2(M(\mathbf{u};\mathbf{w},p),y) = (y - M(\mathbf{u};\mathbf{w},p))^2 | (\mathbf{w},p) \in \Delta_{d-1}\times\mathbb{R} \right\}.
$$

Since both $M(\mathbf{u};\mathbf{w},p)$ and $y_i$ lie within the range $[u_{i(1)}, u_{i(d)}] \subseteq [u_{\min}, u_{\max}]$, we have $y - M(\mathbf{u};\mathbf{w},p) \in [u_{\min} - u_{\max}, u_{\max} - u_{\min}]$. Over the bounded range $[-\gamma,\gamma]$, the function $\ell_2(t,y) = (t-y)^2$ is $2\gamma$-Lipschitz continuous with respect to $t$. Thus, using Talagrand's contraction lemma and Lemma A.6, we get

$$
\hat{\mathfrak{R}}(\mathcal{L}_2) = \hat{\mathfrak{R}}(\ell_2 \circ \mathcal{M}_d) \leq 2(u_{\max} - u_{\min})\hat{\mathfrak{R}}(\mathcal{M}_d).
$$

We then use the uniform convergence bounds for Rademacher complexity to obtain

$$
\sup_{(\mathbf{w},p)\in\Delta_{d-1}\times\mathbb{R}} \left| \hat{R}_n(\mathbf{w},p) - R(\mathbf{w},p) \right| \leq 8(u_{\max} - u_{\min})\hat{\mathfrak{R}}(\mathcal{M}_d) + 3\sqrt{\frac{\log(4/\delta)}{2n}} = \epsilon.
$$

Thus,

$$R(\hat{\mathbf{w}}, \hat{p}) - R(\mathbf{w}, p) = \left( \hat{R}_n(\hat{\mathbf{w}}, \hat{p}) - \hat{R}_n(\mathbf{w}, p) \right) + \left( \hat{R}_n(\mathbf{w}, p) - R(\mathbf{w}, p) \right) + \left( R(\hat{\mathbf{w}}, \hat{p}) - \hat{R}_n(\hat{\mathbf{w}}, \hat{p}) \right)$$

$$\leq 0 + \epsilon + \epsilon = 2\epsilon$$

$$= 16(u_{\max} - u_{\min})\hat{\mathfrak{R}}(\mathcal{M}_d) + 6\sqrt{\frac{\log(4/\delta)}{2n}}.$$

Finally, substituting $\hat{\mathfrak{R}}(\mathcal{M}_d)$ from Lemma A.6 provides the required bounds. $\qquad\square$

## A.6 Proof of Lemma 4.1

First, we state a lemma from Jameson [9]:

**Lemma A.7** (Jameson [9], Theorem 4.6). *Let $f : \mathbb{R} \to \mathbb{R}$ be defined as $f(p) = \sum_{i=1}^{n} a_i \exp(b_i x)$, where $b_1 > b_2 > \ldots > b_n$ and $\sum_{i=1}^{n} a_i = 0$. Define $A_j := \sum_{i=1}^{j} a_i$ and denote by $S(A_j)$ the number of sign changes in the sequence $\{A_i\}_{i=1}^{j}$. Then, the number of unique zeros of $f$ is at most $S(A_n) + 1$.*

Consider the function $f(p) = \sum_{i=1}^{d} w_i u_i^p - \sum_{i=1}^{d} w_i v_i^p$ for $\mathbf{u}, \mathbf{v} \in \mathbb{R}^d$ with disjoint entries:

$$f(p) = \sum_{i=1}^{d} w_i u_i^p - \sum_{i=1}^{d} w_i v_i^p = \sum_{i=1}^{d} w_i \exp(p \log u_i) - \sum_{i=1}^{d} w_i \exp(p \log v_i).$$

Applying Lemma A.7, if $w_i = w$ for all $i$, the sequence $\{A_j\}$, consisting of sums of $w$ or $-w$, can have at most $d-1$ sign changes. A sign change at index $k$ implies $A_{k-1} = 0$, and the next sign change cannot occur before index $k+2$. Therefore, $f(p)$ has at most $d$ zeros in this case. In the general case, where $w_i \neq w_j$ for some $i \neq j$, a sign change in $\{A_j\}$ can occur at any index except the first and the last. Thus, $f(p)$ can have at most $2d-1$ roots, as sign changes are possible at all intermediate indices. We conclude that $M_p(\mathbf{u}; \mathbf{w}, p) - M_p(\mathbf{v}; \mathbf{w}, p)$, defined over $\mathbb{R} \cup \{\pm\infty\}$, can change sign as a function of $p$ at most $d-1$ times if $w_i = \frac{1}{d}$, and up to $2d-1$ times in the general case.

**Lemma A.8.** *Let $r, q : \mathbb{R} \to \mathbb{R}$ be two polynomials such that $c := r(x) - q(x)$ is a constant for all $x \in \mathbb{R}$, and the sets of roots $\{x_1, \ldots, x_d\}$ and $\{y_1, \ldots, y_d\}$ of $r$ and $q$ respectively are disjoint, positive and of size $d$. Then, for $k = 0, 1, \ldots, d-1$, the $k$-th power sums of the roots $x_1, \ldots, x_d$ and $y_1, \ldots, y_d$ are equal, i.e.:*

$$\sum_{i=1}^{d} x_i^k = \sum_{i=1}^{d} y_i^k.$$

*Proof.* Given that $r - q = c$, both polynomials have the same non-constant coefficients. According to Vieta's formulas, the $k$-th elementary symmetric polynomial of $\mathbf{x}$, $e_k(\mathbf{x})$, is the sum of all products of $k$ distinct $x_i$'s, and similarly $e_k(\mathbf{y})$ for $\mathbf{y}$. If $r - q = c$, it implies that the symmetric polynomials derived from the roots of $r$ and $q$ are equal, $e_k(\mathbf{x}) = e_k(\mathbf{y})$.

Newton's identities relate the elementary symmetric polynomials and the power sums as follows:

$$k e_k(\mathbf{x}) = \sum_{i=1}^{k} (-1)^{i-1} e_{k-i}(\mathbf{x}) p_i(\mathbf{x}), \quad k e_k(\mathbf{y}) = \sum_{i=1}^{k} (-1)^{i-1} e_{k-i}(\mathbf{y}) p_i(\mathbf{y}).$$

where $p_i(\mathbf{x})$ is the $i$-power sums of the roots. Given that $e_k(\mathbf{x}) = e_k(\mathbf{y})$, we can equate the right-hand sides of the above identities to obtain the power sums $p_i(\mathbf{x})$ and $p_i(\mathbf{y})$. This yields $p_k(\mathbf{x}) = p_k(\mathbf{y})$ for each $k$, due to the recursive nature of Newton's identities and the fact that the elementary symmetric polynomials of $\mathbf{x}$ and $\mathbf{y}$ are equal for all $k \leq d$.

Hence, $p_k(\mathbf{x}) = p_k(\mathbf{y})$ for all $k = 0, \ldots, d-1$, which concludes the proof. $\qquad\square$

Given $f(p)$ as defined above, Lemma A.8 implies there exists disjoint $\mathbf{u}, \mathbf{v} \in \mathbb{R}^d$ such that $M_p(\mathbf{u}; \mathbf{w}, p) = M_p(\mathbf{v}; \mathbf{w}, p)$, for $d-1$ unique values of $p$ and for $w_i = 1/d$. Moreover, suppose that for a set $\{p_i\}_{i \in [d]}$ there exist $\mathbf{u}, \mathbf{v} \in \mathbb{R}^d$ and $\mathbf{w} \in \Delta_{d-1}$ such that $M_p(\mathbf{u}; \mathbf{w}, p_i) = M_p(\mathbf{v}; \mathbf{w}, p_i)$. Then, for any $\lambda > 0$, there exist $\mathbf{u}', \mathbf{v}' \in \mathbb{R}^d$ such that $M_p(\mathbf{u}'; \mathbf{w}, \lambda p_i) = M_p(\mathbf{v}'; \mathbf{w}, \lambda p_i)$.

**Lemma A.9** (Jameson [9], Theorem 3.4). *For any $k < d$ and $p_1 < \ldots < p_k \in \mathbb{R}$, there exist $\mathbf{u}, \mathbf{v} \in \mathbb{R}_+^d$ and $\mathbf{w} \in \Delta_{d-1}$ such that $M_p(\mathbf{u}; \mathbf{w}, p_i) = M_p(\mathbf{v}; \mathbf{w}, p_i)$ for each $i \le k$. Furthermore, the difference $M_p(\mathbf{u}; \mathbf{w}, p_i) - M_p(\mathbf{v}; \mathbf{w}, p_i)$ does not change sign within any interval $(p_i, p_{i+1})$.*

We now proceed to the proof of Lemma 4.1, which bounds the VC dimensions of the function classes $\mathcal{C}_{\mathbf{w},d}$ and $\mathcal{C}_d$.

*Proof of Lemma 4.1 (a).* As there are at most $2d - 1$ roots to $f(p)$, there can be at most $2d - 1$ sign changes as $p$ varies from $-\infty$ to $\infty$. Consequently, the hypothesis class defined by all $p$ (denoted as $\mathcal{M}_{\mathbf{w},d}$) is a subset of the hypothesis class that consists of at most $2d - 1$ sign changes on the real line. This larger hypothesis class is denoted by $\mathcal{H}_d$, and we have $\mathrm{VC}(\mathcal{M}_{\mathbf{w},d}) \le \mathrm{VC}(\mathcal{H}_d)$.

Let us consider $m$ samples $\{\mathbf{u}_i, \mathbf{v}_i\}_{i=1}^m$. For each sample, sign changes occur at most $2d - 1$ times, and hence the total number of changes in labeling over the entire real line is bounded by $(2d - 1)m$ (as $p$ changes, each change in labeling corresponds to a change in sign for at least one of the samples). This implies that the total number of possible labelings is $(2d - 1)m + 1$.

If the set of $m$ samples is shattered, the upper bound derived above should be at least as large as the total number of labelings possible. We thus have:

$$(2d - 1)m + 1 \ge 2^m.$$

We can show that $m = 2(\lceil \log_2 d \rceil + 1)$ points cannot be shattered. Consider

$$
\begin{aligned}
2^m - (2d-1)m - 1 &= 2^{2(\lceil \log_2 d \rceil + 1)} - 2(2d-1)(\lceil \log_2 d \rceil + 1) - 1 \\
&\ge 2^{2(\log_2 d + 1)} - 4d\lceil \log_2 d \rceil + 2\lceil \log_2 d \rceil - 4d + 1 \\
&= 4d^2 - 4d\lceil \log_2 d \rceil + 2\lceil \log_2 d \rceil - 4d + 1 \\
&= 4d(d - \lceil \log_2 d \rceil) - 4d + 2\lceil \log_2 d \rceil + 1 \\
&\ge 4d - 4d + 2\lceil \log_2 d \rceil + 1, \quad \text{since } d - \lceil \log_2 d \rceil \ge 1 \ \forall d \in \mathbb{N} \\
&= \lceil 2\log_2 d \rceil + 1 > 0.
\end{aligned}
$$

Thus, $m > 2(\log_2 d + 1)$ points cannot be shattered, meaning that $VC(\mathcal{H}_d) < 2(\log_2 d + 1)$. $\quad\square$

We now bound the VC dimension for the unknown weight case. Consider $p \neq 0$. In this case, a hypothesis $C((\mathbf{u}, \mathbf{v}); \mathbf{w}, p)$ can be expressed as

$$
\begin{aligned}
\mathrm{sign}\left( \frac{\log\left(\sum_{i=1}^d w_i u_i^p\right) - \log\left(\sum_{i=1}^d w_i v_i^p\right)}{p} \right) &= \mathrm{sign}\,(p)\,\mathrm{sign}\left(\log\left(\sum_{i=1}^d w_i u_i^p\right) - \log\left(\sum_{i=1}^d w_i v_i^p\right)\right) \\
&= \mathrm{sign}\,(p)\,\mathrm{sign}\left(\left(\sum_{i=1}^d w_i u_i^p\right) - \left(\sum_{i=1}^d w_i v_i^p\right)\right) \\
&= \mathrm{sign}\,(p)\,\mathrm{sign}\,(\langle \mathbf{w}, \mathbf{u}^p - \mathbf{v}^p \rangle) \\
&= \mathrm{sign}\,(\langle \mathbf{w}, \mathrm{sign}\,(p)\,(\mathbf{u}^p - \mathbf{v}^p) \rangle).
\end{aligned}
$$

where $\mathbf{u}^p = (u_1^p \quad \cdots \quad u_d^p)^T$. Thus, for a fixed $p$, the set of viable $\mathbf{w}$'s spans a halfspace. We note that each component of $\mathrm{sign}\,(p)\,(\mathbf{u}^p - \mathbf{v}^p)$ is continuous, which means that $\langle \mathbf{w}, \mathrm{sign}\,(p)\,(\mathbf{u}^p - \mathbf{v}^p) \rangle$ is a continuous function in $\mathbf{w}$ and $p$.

For $n > d$ samples $\{((\mathbf{u}_i, \mathbf{v}_i), y_i)\}_{i=1}^n$, we define $\mathbf{h}_i(p) = \mathrm{sign}\,(p)\,(\mathbf{u}^p - \mathbf{v}^p), i \in [n], p \neq 0$. For a fixed $p$, we note that the set of possible labelings for $\mathbf{w} \in \Delta_{d-1}$ is a subset of the set of possible labelings for $\mathbf{w} \in \mathbb{R}^d$, which in turn is the set of labelings generated by $n$ hyperplanes. Since this problem has VC dimension $d$, the number of possible labelings for a fixed $p$ is upper bounded by $(n+1)^d$. Let $\mathcal{B}(p)$ denote the set of possible labelings for hyperplanes defined by $\{\mathbf{h}_i(p)\}_{i=1}^n$ for a particular $p$.

**Lemma A.10.** *Let $p_1$ and $p_2$ have the same sign, with a labeling $\ell \in \{\pm 1\}^n$ such that $\ell \notin \mathcal{B}(p_1)$ but $\ell \in \mathcal{B}(p_2)$. Then, there is a $p \in [p_1, p_2]$ such that there is a set of $d$ linearly dependent vectors $\mathbf{h}_{(1)}(p), \ldots, \mathbf{h}_{(d)}(p)$.*

*Proof.* Let $\ell$ be the labeling which is in $\mathcal{B}(p_2)$ but not in $\mathcal{B}(p_1)$. Since this labeling is not in $\mathcal{B}(p_1)$, for each $\mathbf{w}$, there is some hyperplane $\mathbf{h}_i(p_1)$ such that $\ell_i \langle \mathbf{w}, \mathbf{h}_i(p_1) \rangle < 0$. Since this labeling is in $\mathcal{B}(p_2)$, there is some $\mathbf{w}$ such that $\ell_i \langle \mathbf{w}, \mathbf{h}_i(p_2) \rangle \geq 0$ for every $i \in [n]$.

Let $\mathfrak{B} \subset \mathbb{R}^d$ denote the unit hypersphere around the origin. Since the labelings are invariant to the scale of $\mathbf{w}$, the set of possible labelings for $\mathbf{w} \in \mathbb{R}^d$ is exactly the set of possible labelings for $\mathbf{w} \in \mathfrak{B}$

Consider the quantity

$$m(p) = \max_{\mathbf{w} \in \mathfrak{B}} \min_i \ell_i \langle \mathbf{w}, \mathbf{h}_i(p) \rangle.$$

We observe that if $m(p) < 0$, for each $\mathbf{w}$ there is some $i \in [n]$ such that $\ell_i \langle \mathbf{w}, \mathbf{h}_i(p) \rangle < 0$, i.e., the labeling is not attained at any point. On the other hand, if $m(p) \geq 0$, there is some $\mathbf{w}$ such that the labeling is attained at $\mathbf{w}$. Since $\ell_i \langle \mathbf{w}, \mathbf{h}_i(p) \rangle$ is a continuous function in $\mathbf{w}$ and $p$ for all $i \in [n]$, $\min_i \ell_i \langle \mathbf{w}, \mathbf{h}_i(p) \rangle$ is also a continuous function in $\mathbf{w}$ and $p$. Thus, $m(p)$ is also a continuous function in $p$.

Using this fact and the intermediate value theorem, there should be some $p \in [p_1, p_2]$ such that $m(p) = 0$. Let $\mathbf{w}^* \in \mathfrak{B}$ be a vector at which $m(p) = 0$ is attained. We now show that at this $p$, at least $d$ of the $n$ vectors $\{\mathbf{h}_i(p)\}_{i=1}^n$ are linearly dependent.

Suppose this were not the case, i.e., any set of $d$ vectors in the set is linearly independent. This means that at most $d - 1$ of the vectors lie on the hyperplane $\{\mathbf{x} : \langle \mathbf{w}, \mathbf{x} \rangle = 0\}$. Let $\mathbf{h}_{(1)}(p), \ldots, \mathbf{h}_{(k)}(p)$ denote these vectors, with $k \leq d - 1$. Since $m(p) = 0$, there should be at least one such vector. Let $\mathbf{H} \in \mathbb{R}^{k \times d}$ be the matrix with these vectors as the rows.

If these $k$ vectors are linearly dependent, we can add any of the remaining $n - k$ vectors to get a set of $d$ linearly dependent vectors. Let us consider the case where they are not linearly dependent. Consider $\{\mathbf{x} : \mathbf{H}\mathbf{x} = \mathbf{1}_k\}$. This is an underdetermined set of linear equations, and the set should be non-empty (because of linear independence of the $k$ vectors). Let $\mathbf{x}_0$ be one of the vectors in this set.

Let $t > \max_{i \in [n]} -\frac{\langle \mathbf{w}, \mathbf{h}_i(p) \rangle}{\langle \mathbf{x}_0, \mathbf{h}_i(p) \rangle}$. We have

$$
\begin{aligned}
\langle \mathbf{w} + t\mathbf{x}_0, h_i(p) \rangle &= \langle \mathbf{w}, h_i(p) \rangle + t \langle \mathbf{x}_0, h_i(p) \rangle \\
&> \langle \mathbf{w}, h_i(p) \rangle - \max_{j \in [n]} \frac{\langle \mathbf{w}, \mathbf{h}_j(p) \rangle}{\langle \mathbf{x}_0, \mathbf{h}_j(p) \rangle} \langle \mathbf{x}_0, \mathbf{h}_i(p) \rangle \\
&\geq \langle \mathbf{w}, h_i(p) \rangle - \frac{\langle \mathbf{w}, \mathbf{h}_i(p) \rangle}{\langle \mathbf{x}_0, \mathbf{h}_i(p) \rangle} \langle \mathbf{x}_0, \mathbf{h}_i(p) \rangle \\
&= 0.
\end{aligned}
$$

Thus, $\mathbf{w} + t\mathbf{x}_0$ is a point such that $\langle \mathbf{w} + t\mathbf{x}_0, \mathbf{h}_i(p) \rangle > 0$ for all $i \in [n]$. This means that $m(p) > 0$, which is a contradiction. Intuitively, this means that if any $d$ vectors in $\{h_i(p)\}_{i=1}^n$ are linearly independent, then $m(p) > 0$. Thus, there should be a set of $d$ linearly dependent vectors $h_{(1)}(p), \ldots, h_{(d)}(p)$. $\qquad\square$

From the above lemma, we observe that any change in the set of labelings is accompanied by a $p$ which gives $d$ linearly dependent vectors.

*Proof of Lemma 4.1 (b).* Using the lemma above, to bound the number of possible labelings, we first bound the number of $p$'s such that there are $d$ linearly dependent vectors.

Consider a set of $d$ vectors $h_1(p), \ldots, h_d(p)$. As the vectors are linearly dependent, the determinant of the matrix constructed using these vectors should be zero. We should thus have

$$
\begin{vmatrix}
\text{sign}(p)(u_{11}^p - v_{11}^p) & \cdots & \text{sign}(p)(u_{1d}^p - v_{1d}^p) \\
\vdots & \ddots & \vdots \\
\text{sign}(p)(u_{d1}^p - v_{11}^p) & \cdots & \text{sign}(p)(u_{1d}^p - v_{dd}^p)
\end{vmatrix} = 0.
$$

Upon expanding the determinant, we get an equation of the form $\sum_{i=1}^m a_i u_i^p$ which has $2^d \cdot d!$ terms. From an earlier lemma, we know that this equation should have at most $2^d \cdot d! - 1$ roots. Upon adding the original configuration, we get $2^d \cdot d!$ possible configurations. The choice of the $d$ vectors can be

made in $\binom{n}{d}$ ways, and hence we have a bound on the possible changes as $2^d d! \binom{n}{d}$. In the worst case, we assume that all the labelings are changed, and we thus get an upper bound on the changes as

$$(n+1)^d 2^d d! \binom{n}{d}.$$

In the beginning of the proof, we had carefully set aside $p = 0$. We now observe that $p = 0$ is a root of the above system of equations. Thus, we are implicitly considering any possible changes at $p = 0$ as well.

We can show that $n = 8(\lceil d \log_2 d \rceil + 1)$ points cannot be shattered.

$$(n+1)^d 2^d d! \binom{n}{d} = (n+1)^d 2^d \cdot n(n-1)\dots(n-d+1)$$
$$> (n+1) 2^d n^{2d-1}$$
$$> 2^{d-1} n^{2d}.$$

We now show that for every $d \in \mathbb{N}$, the inequality $2^{d-1} n^{2d} \leq 2^n$ holds, i.e. $n - d - 2d \log_2 n + 1 > 0$ for $n = 8(\lceil d \log_2 d \rceil + 1)$. For $d \in \{1, 2\}$, this statement can be verified directly. Therefore, it suffices to show that $f(x) := 8(x \log_2 x + 1) - 2x \log_2[8(x \log_2 x + 1)] - x + 1 > 0$ for any $x \geq 3$.

$$f(x) > 8x \log_2 x - 2x \log_2(16x \log_2 x) - x + 9, \quad \text{for } x \log_2 x > 1,$$
$$= 6x \log_2 x - 2x \log_2(\log_2 x) - 9x + 9$$
$$> 4x \log_2 x - 9x + 9 = g(x).$$

We note that $g(3) = 12 \log_2 3 - 18 > 1 > 0$. Moreover, $g'(x) = 4 \log_2 x + 4/\log 2 - 9$, and we note that $g'(2) = 4/\log 2 - 5 > 0.77 > 0$, with $g'(x)$ being an increasing function. Thus, $f(x) > g(x) > 0$ for all $x \geq 3$. This means that $f(x) > 0$, which proves our bound. This implies that the VC dimension is bounded above by $8(d \log_2 d + 1)$ □

*Proof of Lemma 4.1 (c).* Take $m = \log_2(d) + 1$. Let $\sigma_1, \dots, \sigma_{2^m} \in \{\pm 1\}^m$ be a Gray code ordering of the set $\{\pm 1\}^m$, such that two successive values have a Hamming distance of 1, and that the number of changes in different bit positions is at most $\frac{2^m}{2} \leq d$ (for the existence of such an ordering, see [2]). For $i < 2^m$, denote by $s_i \in [m]$ the bit position in which $\sigma_i$ and $\sigma_{i+1}$ differ. By using Lemma A.9 $d$ times, there exists a sample $\{\mathbf{u}_i, \mathbf{v}_i\}_{i=1}^m$ and $p_1 < \dots < p_{2^m-1}$, where $p_i$ satisfies $\|\mathbf{u}_j\|_{p_i} = \|\mathbf{v}_j\|_{p_i}$ if $s_i = j$. Furthermore, define $p_0 = -\infty$, and note that each interval $(p_i, p_{i+1})$ for $0 \leq i < 2^m$ corresponds to a unique combination of labels over $\{\mathbf{u}_i, \mathbf{v}_i\}_{i=1}^m$.

When the weights are unknown, we observe that $p = 1$ is similar to the case of linear classification with the constraint of positive weights and no bias term. This is because

$$C((\mathbf{u}, \mathbf{v}); \mathbf{w}, 1) = \text{sign}\left(\log M(\mathbf{u}; \mathbf{w}, 1) - \log M(\mathbf{v}, 1)\right)$$
$$= \text{sign}\left(M(\mathbf{u}; \mathbf{w}, 1) - M(\mathbf{v}; \mathbf{w}, 1)\right)$$
$$= \text{sign}\left(\sum_{i=1}^d w_i(u_i - v_i)\right).$$

Since linear classification with no bias term has a VC dimension of $d - 1$, this is a lower bound for the VC dimension of $\mathcal{C}_{\mathbf{w}, d}$. □

### A.7 Proof of Theorem 4.3

*Proof.* We prove the result for unknown weights, with the known weights result following similar steps. We consider the function class $\mathcal{C}_d$ as in Section 4, with $\ell_{0-1}$ loss being $\ell_{0-1}(t, y) = (1 + ty)/2, t, y \in \{\pm 1\}$. We observe that $\ell_{0-1}$ is $1/2$ Lipschitz w.r.t. $t$. Thus, by applying Theorem 3 of Natarajan et al. [16], we observe that w.r.t. $\ell_{0-1}$ on the *noiseless* data distribution,

$$R(\hat{\mathbf{w}}, \hat{p}) - R(\mathbf{w}, p) \leq 4L_\rho \hat{\mathfrak{R}}(\mathcal{C}_d) + 2\sqrt{\frac{\log(1/\delta)}{2n}} \tag{2}$$

where $L_\rho = (1 + |\rho_{+1} - \rho_{-1}|)L/(1 - \rho_{+1} - \rho_{-1})$. Here, $\rho_{+1}$ and $\rho_{-1}$ are defined as the probability of mislabeling true positive and true negative examples, which in our case are the same value, $\rho$. Thus, $L_\rho = 1/(2(1 - 2\rho))$ in our case. We obtain $\hat{\mathfrak{R}}(\mathcal{C}_d)$ using the VC bound on Rademacher complexity:

$$\hat{\mathfrak{R}}(\mathcal{C}_d) \leq \sqrt{\frac{16(d\log_2 d + 1)\log(n+1)}{n}}.$$

Substituting it in Equation (2) concludes our proof. $\square$

## A.8 Proof of Theorem 5.1

*Proof.* We prove the result for the case of unknown weights; the known weights case follows similar steps. First, we establish a bound on the Rademacher complexity of

$$\mathcal{T}_{\mathbf{w},d} = \{\tau\left(\log M(\cdot; \mathbf{w}, p) - \log M(\cdot; \mathbf{w}, p)\right) \mid \tau, p\}.$$

Define

$$\log M(\mathbf{u}_i; \mathbf{w}, p) = \log u_{i(1)} + \log\left(\frac{u_{i(d)}}{u_{i(1)}}\right)\log M(\mathbf{r}_i; \mathbf{w}, q),$$

and

$$\log M(\mathbf{v}_i; \mathbf{w}, p) = \log v_{i(1)} + \log\left(\frac{v_{i(d)}}{v_{i(1)}}\right)\log M(\mathbf{r}'_i; \mathbf{w}, q).$$

$$\hat{\mathfrak{R}}(\mathcal{T}_d) = \frac{1}{n}\mathbb{E}_\epsilon\left[\sup_{\tau,\mathbf{w},p}\sum_{i=1}^n \epsilon_i\tau\left(\log M(\mathbf{u}_i; \mathbf{w}, p) - \log M(\mathbf{v}_i; \mathbf{w}, p)\right)\right]$$

$$\leq \frac{1}{n}\mathbb{E}_\epsilon\left[\sup_{\tau,\mathbf{w},p}\sum_{i=1}^n \epsilon_i\tau\log M(\mathbf{u}_i; \mathbf{w}, p)\right] + \frac{1}{n}\mathbb{E}_\epsilon\left[\sup_{\tau,\mathbf{w},p}\sum_{i=1}^n (-\epsilon_i)\tau\log M(\mathbf{v}_i; \mathbf{w}, p)\right]$$

$$\leq \frac{1}{n}\mathbb{E}_\epsilon\left[\sup_{\tau,\mathbf{w},p}\sum_{i=1}^n \epsilon_i\tau\log M(\mathbf{u}_i; \mathbf{w}, p)\right] + \frac{1}{n}\mathbb{E}_\epsilon\left[\sup_{\tau,\mathbf{w},p}\sum_{i=1}^n \epsilon_i\tau\log M(\mathbf{v}_i; \mathbf{w}, p)\right]$$

$$\leq \frac{1}{n}\mathbb{E}_\epsilon\left[\sup_{\tau,\mathbf{w},p}\sum_{i=1}^n \epsilon_i\tau\left(\log u_{i(1)} + \log\left(\frac{u_{i(d)}}{u_{i(1)}}\right)\log M(\mathbf{r}_i; \mathbf{w}, q)\right)\right]$$

$$\quad + \frac{1}{n}\mathbb{E}_\epsilon\left[\sup_{\tau,\mathbf{w},p}\sum_{i=1}^n \epsilon_i\tau\left(\log v_{i(1)} + \log\left(\frac{v_{i(d)}}{v_{i(1)}}\right)\log M(\mathbf{r}'_i; \mathbf{w}, q)\right)\right]$$

$$= \frac{1}{n}\mathbb{E}_\epsilon\left[\sup_{\tau,\mathbf{w},p}\sum_{i=1}^n \epsilon_i\tau\log\left(\frac{u_{i(d)}}{u_{i(1)}}\right)\log M(\mathbf{r}_i; \mathbf{w}, q)\right]$$

$$\quad + \frac{1}{n}\mathbb{E}_\epsilon\left[\sup_{\tau,\mathbf{w},p}\sum_{i=1}^n \epsilon_i\tau\log\left(\frac{v_{i(d)}}{v_{i(1)}}\right)\log M(\mathbf{r}'_i; \mathbf{w}, q)\right]$$

$$\leq \frac{\kappa}{n}\mathbb{E}_\epsilon\left[\sup_{\tau,\mathbf{w},p}\left|\sum_{i=1}^n \epsilon_i\tau\log M(\mathbf{r}_i; \mathbf{w}, q)\right|\right] + \frac{\kappa}{n}\mathbb{E}_\epsilon\left[\sup_{\tau,\mathbf{w},p}\left|\sum_{i=1}^n \epsilon_i\tau\log M(\mathbf{r}'_i; \mathbf{w}, q)\right|\right]$$

$$\leq \frac{\tau_{\max}\kappa}{n}\mathbb{E}_\epsilon\left[\sup_{\tau,\mathbf{w},p}\left|\sum_{i=1}^n \epsilon_i\log M(\mathbf{r}_i; \mathbf{w}, q)\right|\right] + \frac{\tau_{\max}\kappa}{n}\mathbb{E}_\epsilon\left[\sup_{\tau,\mathbf{w},p}\left|\sum_{i=1}^n \epsilon_i\log M(\mathbf{r}'_i; \mathbf{w}, q)\right|\right]$$

$$= 2\tau_{\max}\kappa\hat{\mathfrak{R}}_{\text{abs}}(\mathcal{S}_d).$$

We now apply the bound on $\hat{\mathfrak{R}}_{\text{abs}}(\mathcal{S}_d)$ from Appendix A.4 to obtain the following bound on $\hat{\mathfrak{R}}(\mathcal{T}_d)$:

$$\hat{\mathfrak{R}}(\mathcal{T}_d) \leq 2\tau_{\max}\kappa\left(\sqrt{\frac{2\log 2 + 16(d\log_2 d + 1)\log n}{n}} + \frac{c}{\sqrt{n}}\right).$$

Finally, we use the uniform convergence bounds derived from Rademacher complexity to establish the following PAC bound:

$$R(\hat{\mathbf{w}}, \hat{p}) - R(\mathbf{w}, p) = \left( \hat{R}_n(\hat{\mathbf{w}}, \hat{p}) - \hat{R}_n(\mathbf{w}, p) \right) + \left( \hat{R}_n(\mathbf{w}, p) - R(\mathbf{w}, p) \right) + \left( R(\hat{\mathbf{w}}, \hat{p}) - \hat{R}_n(\hat{\mathbf{w}}, \hat{p}) \right)$$

$$\leq 0 + \epsilon + \epsilon = 2\epsilon$$

$$= 8\kappa \hat{\mathfrak{R}}(\mathcal{T}_d) + 6\sqrt{\frac{\log(4/\delta)}{2n}}.$$

$\square$

## B   Algorithm

Our algorithm can be broken into two nested steps. The first step consists of choosing $p$, and the second step involves conducting gradient descent on $\mathbf{w}$ (and possibly $\tau$) to obtain their empirically optimal values, $\hat{\mathbf{w}}$ and $\hat{p}$. In our experiments we choose $p$ using grid search. However, optimization over $p$ can also be done using other methods like simulated annealing. We minimize the $\ell_2$ loss in the cardinal case with weighted power mean and the logistic loss in the ordinal case with log weighted power mean. The algorithm's pseudocode is presented in Algorithm 1.

---

**Algorithm 1** ERM algorithm for weighted power mean-based optimization

---

**Require:** $\mathcal{D} = \{(\mathbf{x}_i, y_i)\}_{i=1}^n$
  $\hat{\mathbf{w}} \leftarrow \mathbf{1}/d$
  $v_{\text{best}} \leftarrow 0$
  $\hat{p} \leftarrow 0$
  **for** $p \in [p_{\text{lower}}, p_{\text{lower}} + \epsilon, \ldots, p_{\text{upper}} - \epsilon, p_{\text{upper}}]$ **do**
    $v \leftarrow \arg\min_{\mathbf{w}} \frac{1}{n} \sum_{i=1}^n \ell(M(u_i; \mathbf{w}, p), y_i)$
    $\tilde{\mathbf{w}} \leftarrow \arg\min_{\mathbf{w}} \frac{1}{n} \sum_{i=1}^n \ell(M(u_i; \mathbf{w}, p), y_i)$
    **if** $v < v_{\text{best}}$ **then**
      $\hat{\mathbf{w}} \leftarrow \tilde{\mathbf{w}}$
      $v_{\text{best}} \leftarrow v$
    **end if**
  **end for**
  **Return** $\hat{\mathbf{w}}$, $v_{\text{best}}$

---

Note that for the ordinal case, we would optimize over $\tau$ along with $\mathbf{w}$. For our experiments, we set $p_{\text{lower}} = -3.5$ and $p_{\text{upper}} = 3.5$. We use a grid resolution of $\epsilon = 0.1$. Since the function is not convex, we use several tricks to ensure quick convergence:

- While we use Algorithm 1 from [5] for projection onto the simplex, it can potentially be time consuming. Thus, we project the gradient $\nabla_{\mathbf{w}} \ell$ itself on the unit simplex and use it for gradient descent, with the simplex projection algorithm being used only when some weights become too small/negative.

- To prevent the algorithm from taking excessively large steps, we use the learning rate to clip the norm of the gradient. More specifically, if $g_t$ is the gradient and $\lambda$ is the learning rate, we use the update

$$g_{t+1} = g_t - \min\{\lambda, \|g_t\|_2\} \cdot \frac{g_t}{\|g_t\|_2}.$$

- If the optimal value hasn't improved in a certain number of iterations, the algorithm may be oscillating above the minimum. We thus halve the learning rate to encourage better convergence.

- If the learning rate becomes too small, the steps taken would be too small to change the loss significantly. Thus, we terminate the algorithm. We also terminate the algorithm if the range of the past few losses is too small.

- We also conduct gradient descent parallely starting from $d + 1$ points. The $d + 1$ points correspond to points close to the vertices of the simplex (corresponding to almost one-hot vectors) and the centroid of the simplex. This was done since convergence was observed to be slow for certain weights. At each step, $v_{\text{best}}$ is updated according to the point giving the minimum loss.

We ran the experiments on an NVIDIA RTX A5000 GPU. The algorithm with the above settings takes about 30 minutes to check for all 71 values of $p$.

## C  Semi-synthetic Experiments: Further Information

Here we provide additional results and details from our semi-synthetic experiments. Figure 3 shows cardinal case results with varying sample sizes and noise levels. Figure 4 provides ordinal case results across different sample sizes and values of $\tau$. Figure 5 displays ordinal case results for $p = 1.62$ across varying values of $\tau$.

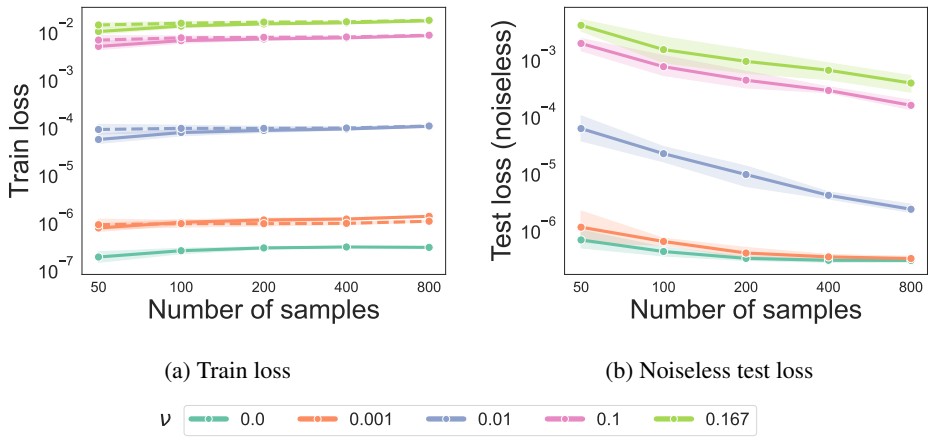

(a) Train loss                        (b) Noiseless test loss

$\nu$ ━●━ 0.0   ━●━ 0.001   ━●━ 0.01   ━●━ 0.1   ━●━ 0.167

Figure 3: More results for cardinal case with number of samples. Different lines show results for different values of added noise. Solid lines correspond to values for learnt parameters, whereas dotted lines correspond to values for real parameters.

## D  Additional Plots

Figure 6 shows a pair of utility vectors $(\mathbf{u}, \mathbf{v})$ such that with $\mathbf{w} = \mathbf{1}_d/d$, $\log M(\mathbf{u}; \mathbf{w}, p) - \log M(\mathbf{v}; \mathbf{w}, p)$ is non-convex. Upon slightly changing the value of $\mathbf{v}$ to $\mathbf{v}'$, we see that there can be significant change in the region $\{p : \log M(\mathbf{u}; \mathbf{w}, p) - \log M(\mathbf{v}; \mathbf{w}, p) > 0\}$.

## E  Simulations

We conduct additional simulations on cardinal and ordinal data with logistic noise.

For each $d$ and $n$, we construct a dataset in a specified range $[u_{\min}, u_{\max}]^d = [1, 1000]^d$. Each individual $i$ is assumed to have a scaled and translated beta distribution over $[u_{\min}, u_{\max}]$, with the parameters $(\alpha_i, \beta_i)$ differing for each $i$. Utilities for each action are drawn independently for each individual to construct a utility vector. The underlying weight vector is sampled uniformly from $\Delta_{d-1}$.

To learn $p$ (and $\mathbf{w}$ if needed), we assume $p$ to be in a fixed range, in this case $[-10, 10]$. We begin with a random sampling stage, where $N_{\text{random}}$ instances of $p$ (and $\mathbf{w}$) are uniformly sampled. At the end of this stage, we select the parameter set with the lowest training loss and then perform gradient descent for $N_{\text{grad}}$ steps. We observe that this two-stage method yields good results across the values of $d$ we consider. Each setting is run three times to obtain error bounds on the empirical results.

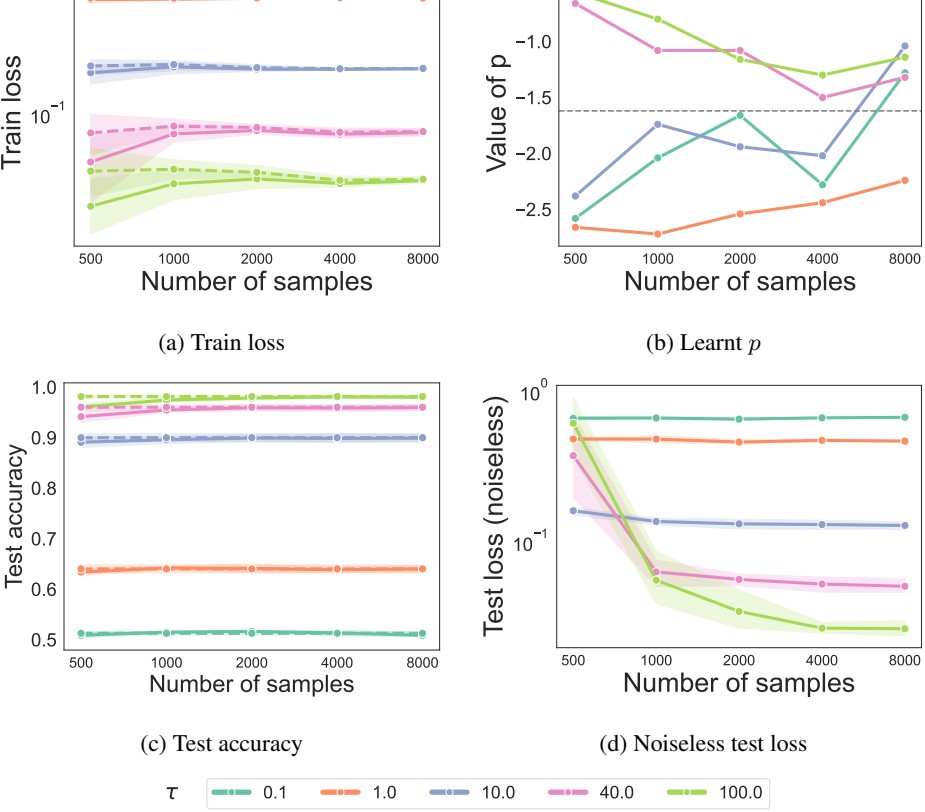

(a) Train loss                                          (b) Learnt $p$

(c) Test accuracy                                       (d) Noiseless test loss

$\tau$    —•— 0.1    —•— 1.0    —•— 10.0    —•— 40.0    —•— 100.0

Figure 4: More results for ordinal case with number of samples. Different lines show results for different values of $\tau$. Solid lines correspond to values for learnt parameters, whereas dotted lines correspond to values for real parameters.

In the unknown weights case, we observe that sampling occurs in $d$ dimensions. As $d$ increases, we encounter the curse of dimensionality, meaning that $N_{\text{random}}$ would need to grow exponentially with $d$ to maintain sampling density across different $d$. This makes maintaining the same density impractical for larger dimensions. As a compromise, we increase $N_{\text{random}}$ linearly with $d$.

### E.1 Cardinal Values

For cardinal values, we add Gaussian noise to each $y_i$ with standard deviation $(u_{(d)} - u_{(1)})/10$ and clamp the values to $[u_{(1)}, u_{(d)}]$. Experiments are conducted for both known and unknown weights with $p = -2$. Figure 7a (known weights) and Figure 7b (unknown weights) show the estimated test loss on noiseless test data generated using the true parameters.

We observe relatively little change in the test loss difference for known weights as $n$ increases. However, for higher $d$, there is a greater decrease in test loss with increasing $n$ when weights are also being learned. The estimated test loss also increases with $d$, with a stronger trend for unknown weights.

### E.2 Logistic Noise

For logistic noise, we generate pairs of utility vectors with $p = 0.9$ and a $\mathbf{w}$ obtained through random sampling, then mislabel each instance according to Equation (1) with $\tau^* = 10$. Since we also need to learn $\tau$, we set $\tau_{\max} = 50$ and sample it uniformly along with $p$ (and $\mathbf{w}$). Figure 7a (known weights) and Figure 7b (unknown weights) show the accuracy on noiseless test data of the learned parameters.

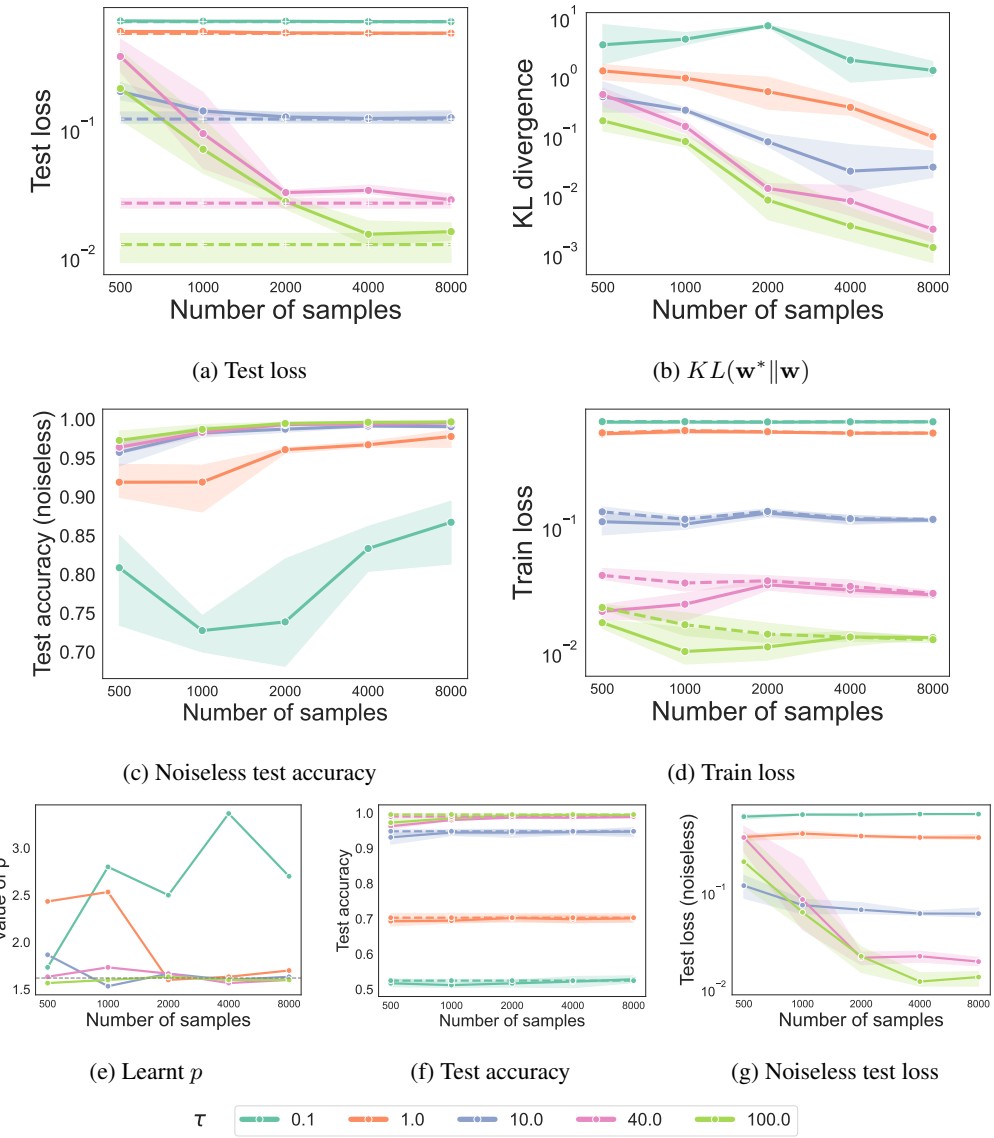

Figure 5: More results for ordinal case with $p = 1.62$. Different lines show results for different values of $\tau$. Solid lines correspond to values for learnt parameters, whereas dotted lines correspond to values for real parameters.

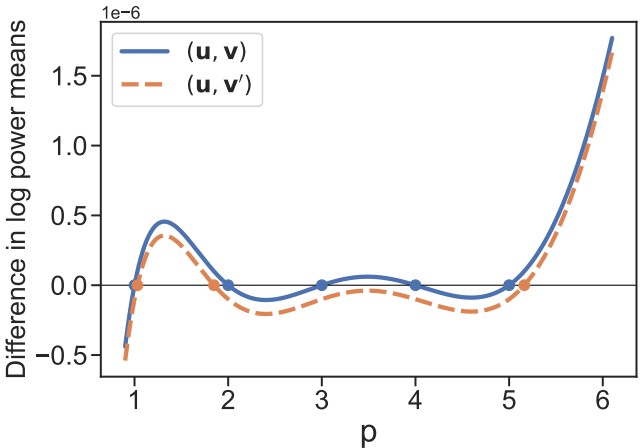

Figure 6: An example showing the non-convexity of $\log M(\mathbf{u}, \mathbf{w}, p) - \log M(\mathbf{v}, \mathbf{w}, p)$. We see that the function has five roots for $(\mathbf{u}, \mathbf{v})$, but is translated downwards for $(\mathbf{u}, \mathbf{v}')$ and has only three roots in this case. If the correct label is 1 for both pairs, then $p$ should be greater than 6; however, gradient-based optimization can stop between 3 and 4, which is a local optimum and does not give correct labels to both points.

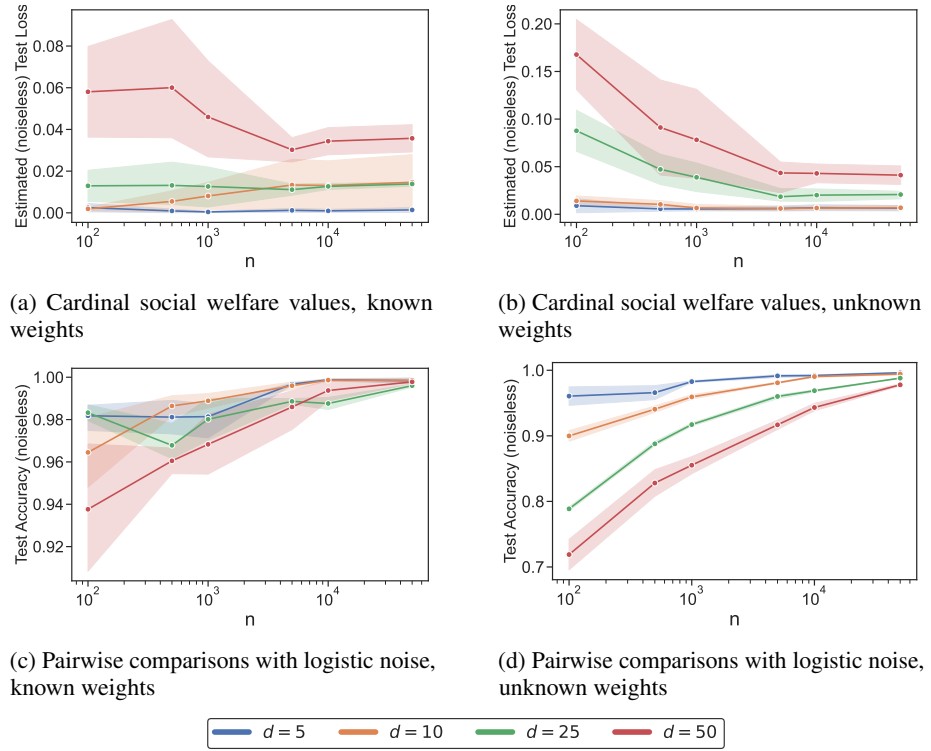

(a) Cardinal social welfare values, known weights

(b) Cardinal social welfare values, unknown weights

(c) Pairwise comparisons with logistic noise, known weights

(d) Pairwise comparisons with logistic noise, unknown weights

Figure 7: Results for synthetic data on cardinal and ordinal logistic tasks

Across different settings, the proportion of correctly labeled samples in the training dataset has a mean of $71.4\%$, with a maximum of $86.5\%$.

For known weights, accuracy increases with $n$, and mean accuracy remains high ($> 93\%$) across $d$. Differences between curves for various values of $d$ are minimal, with all approaching near-perfect accuracy as $n$ becomes large. This suggests that error bounds may be independent of $d$. For unknown weights, a clear trend of decreasing performance with increasing $d$ is observed, expected due to the $\mathcal{O}(\sqrt{d \log d})$ dependence of logistic loss error bounds. Nevertheless, all settings achieve high

accuracy as $n$ increases. Up to moderately high $d$, the logistic noise model successfully finds highly accurate parameters despite significant mislabeling in the training data.

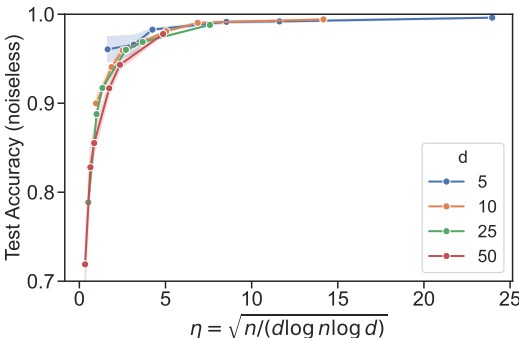

Figure 8: Verification of $\mathcal{O}(d \log d)$ risk bound for ordinal case with logistic noise, unknown weights

In Figure 8, we re-plot the test accuracy $\alpha$ on noiseless data against $\eta = \sqrt{n/(d \log n \log d)}$, a re-scaled version of Figure 7d. Theoretically, $\alpha$ and $\eta$ are related as $1 - \alpha = \mathcal{O}(1/\eta)$. The alignment of all curves in Figure 8, compared to the original curves in Figure 7d, provides evidence that our risk and sample complexity bounds indeed scale as $d \log n \log d$ for the ordinal case with logistic noise and unknown weights.

