# OpenReview forum: "Learning Social Welfare Functions"
_NeurIPS.cc/2024/Conference — NeurIPS 2024 spotlight_

### Official Review · Reviewer_1QjM · 2024-06-30

**Soundness:** 4
**Presentation:** 4
**Contribution:** 4
**Rating:** 7
**Confidence:** 4

**Summary:**

The authors study the learnability of social welfare functions given decisions data by a central decision-maker that is taking into account their constituents' welfare. They discuss PAC bounds according to a number of settings with a focus on weighted power mean functions. These settings include cardinal utility vectors under a target social welfare functions and pairwise comparisons between utility vectors. Learning is taken either without noise, with iid noise, or logistic noise. The authors validate their theoretical findings by learning welfare functions on proprietary data by Lee et. al. (2018).

**Strengths:**

Interesting novel concept and quality results. This was a pleasure to read and will be a useful contribution to the research community.

**Weaknesses:**

I'd be interested if you could discuss some implications or interpretations of your work. You obtain PAC bounds for the various settings and you summarize your results in Table 1. I am not immediately sure what the quality of these results are, how they compare to prior work, and how they would be represented in real-world learning.

Minor:
- Perhaps include a short primer on VC dimension, pseudo-dimension, Rademacher complexity, and PAC learning for unfamiliar audiences in the appendix
- Perhaps cite (Xia, AAMAS 2013) and related papers on preference/rank learning (e.g., (Zhao, Liu, and Xia, IJCAI 2022) or (Newman, Royal Society 2022) or (Conitzer and Sandholm, UAI 2005) or (Xia, Conitzer, and Lang, AAMAS 2010)) as related work
- Line 142: "welfare" not "malfare"

**Questions:**

NA

**Limitations:**

Yes

---

> ### Author Rebuttal · Authors · 2024-08-06
>
> > I'd be interested if you could discuss some implications or interpretations of your work. You obtain PAC bounds for the various settings and you summarize your results in Table 1. I am not immediately sure what the quality of these results are, how they compare to prior work, and how they would be represented in real-world learning.
>
> Our PAC bounds provide theoretical guarantees on the sample complexity of learning social welfare functions, demonstrating that accurate learning is possible with a reasonable number of samples. These bounds are the first of their kind, as prior work [1] focused on more restricted classes and did not provide finite-sample guarantees. There is plethora of other work on preference learning and ranking, including the papers the reviewer suggested, but these don't focus on learning social welfare functions to the best of our knowledge. We showcase the practical feasibility of our approach on a real-world dataset of food allocation decisions [2], highlighting its potential to uncover valuable insights about decision-making priorities. Learning social welfare functions from data can significantly impact society by enabling us to understand a decision maker's priorities and notions of fairness, ultimately helping to identify biases and inform the design of improved policies. However, we recognize the crucial ethical considerations surrounding the deployment of such learning systems, particularly the risks associated with model misspecification. Our focus on the axiomatically-justified family of weighted power means mitigates these risks, but careful validation remains essential.
> We will incorporate a more detailed discussion of these implications and expand on the connections to related work on preference learning and decision theory in the revised paper.
>
> > Perhaps include a short primer on VC dimension, pseudo-dimension, Rademacher complexity, and PAC learning for unfamiliar audiences in the appendix.
>
> We will add a primer on key learning-theoretic concepts to the appendix, making the paper more accessible to a broader audience.
>
> > Perhaps cite (Xia, AAMAS 2013) and related papers on preference/rank learning (e.g., (Zhao, Liu, and Xia, IJCAI 2022) or (Newman, Royal Society 2022) or (Conitzer and Sandholm, UAI 2005) or (Xia, Conitzer, and Lang, AAMAS 2010)) as related work
>
> Thank you for suggesting these papers, which we're familiar with; we're happy to discuss them. Here's a quick comparison:
>
> [3] provides a high-level overview of methods for designing novel social choice mechanisms. It identifies challenges and develops a three-stage workflow to combine machine learning methods and social choice axioms. While their paper provides prescriptions for learning social choice functions, they do not study the complexity of learning these functions from data. Our paper focuses on a particular class of social *welfare* functions and demonstrates polynomial sample complexity and efficient learnability in practice.
>
> [4] studies random utility models (RUMs) with 3 modifications, with a special focus on Plackett-Luce (PL) models. It presents a sufficient condition and an EM algorithm for MLE. This work concerns preference learning and models individual utilities for agents. By contrast, our work models how these individual utilities are combined to generate social welfare, and this problem has a different axiomatic basis which gives rise to a different function family. We note that preference learning and social welfare learning can be a part of the same pipeline, with the former modeling utilities and the latter using these utilities to model social welfare.
>
> [5] addresses the problem of ranking entities through pairwise comparisons, where comparisons can contradict each other. It gives an EM algorithm using a BTL model modified to capture different types of comparisons. However, this work develops a ranking for the current set of alternatives, whereas our goal is to learn the social welfare function, which can be used for completely new comparisons in the ordinal case.
>
> [6] explores which common voting rules have a noise model which makes the rule an MLE, establishing results for various popular voting rules. [7] extends this work further by considering aggregation of multi-issue voting, using CP-nets to represent how issues depend on each other. Both papers mention two perspectives for social choice: 1) agents' diverse preferences are the basis for joint decisions, 2) there are correct joint decisions with agents having different perceptions, and thus each agent has a noisy version of this correct joint decision. While both papers adopt the second perspective, our work is aligned with the first.
>
> [1] A. D. Procaccia, A. Zohar, Y. Peleg, and J. S. Rosenschein. The learnability of voting rules. Artificial Intelligence, 2009.
>
> [2] Min Kyung Lee, et al. WeBuildAI: Participatory framework for algorithmic governance. CSCW 2019.
>
> [3] Xia, Lirong. Designing social choice mechanisms using machine learning. AAMAS 2013.
>
> [4] Zhao, Z., Liu, A., & Xia, L. Learning mixtures of random utility models with features from incomplete preferences. IJCAI 2022.
>
> [5] Newman, M. E. Ranking with multiple types of pairwise comparisons. Proceedings of the Royal Society A, 2022.
>
> [6] Conitzer, V., & Sandholm, T. Common voting rules as maximum likelihood estimators. UAI 2005.
>
> [7] Xia, Lirong, Vincent Conitzer, and Jérôme Lang. Aggregating preferences in multi-issue domains by using maximum likelihood estimators. AAMAS 2010.

---

> > ### Comment · Reviewer_1QjM · 2024-08-10
> > **Thank you for your response.**
> >
> > Thank you for your response.

---

### Official Review · Reviewer_W5KA · 2024-07-06

**Soundness:** 3
**Presentation:** 2
**Contribution:** 2
**Rating:** 4
**Confidence:** 1

**Summary:**

This paper studies the learnability of social welfare functions -- which are functions over the utility of a group of voters and an outcome. Under varying information schemes, they address the question of how well it is possible to learn the social welfare function being used by a decision maker. The first setting considers learning when cardinal values of actions are knowable, which the authors point out corresponds to regression. The second setting looks at when the information given is a pair of utilitty vectors and some indication of which vector corresponds to higher social welfare. Finally, the authors consider the pairwise model when information is noisy. The paper shows that in all settings being considered, a large class of social welfare functions are learnable with a polynomial number of samples. Experiments demonstrate the existence of a practical algorithm for these results.

**Strengths:**

The problem being studied in this paper is well-defined and seems like it may be interesting. Despite the questions asked not being obscure, I am aware of little work that studies similar ideas ([17] being the exception, and I've always thought it odd that more work has not directly built atop [17]).

The methodology taken in the paper seems quite reasonable. While the proofs are not included in the body the results appear correct, to my limited understanding.

This problem is certainly able to inspire potential future research and provides a reasonable contribution in its own right.

**Weaknesses:**

While only incidentally a weakness of this particular paper, the state of science would be better if this were three separate papers (or a journal submission). There is simply not enough time to consider the paper in depth and the parts of the paper important for peer-review (ie. the proofs and many validation experiments) are not in the part of the paper that gets the bulk of reviewing attention so my understanding of the paper is quite limited; I found the math quite dense and open to improved clarity.

As far as clarity is concerned, the paper is moderately readable but I feel that the results could be explained somewhat more clearly without requiring additional space. The introduction does an adequate job of outlining the common idea of social welfare and what problem the paper studies. I understand what the problem solves but not until the end of Section 7 is there some suggestion of why this problem might be interesting. Motivating the questions in the paper earlier would be useful.

**Questions:**

N/A

**Limitations:**

Some discussion has been included but it is somewhat limited and surface level.

---

> ### Author Rebuttal · Authors · 2024-08-06
>
> At the risk of slightly abusing the rebuttal, we note that overall you seem to have a positive view of the paper. You write that the problem "seems like it may be interesting" and it "provides an interesting contribution in its own right," the methodology "seems quite reasonable," the results "appear to be correct," and the work "is certainly able to inspire potential future research." The weaknesses you list don't appear to be criticisms of the paper; the first seems to reflect a shortcoming of the NeurIPS reviewing timeline, and the second is that the paper is "moderately readable" with a suggestion (which we would be happy to follow) to move some of the discussion to the introduction.
>
> In light of this, we would be very grateful if you would consider updating your rating of the paper. Alternatively, we would kindly ask that you clarify the weaknesses that lead you to recommend rejection.

---

> > ### Comment · Reviewer_W5KA · 2024-08-12
> >
> > TL;DR: The math/theory is not explained clearly enough for someone not already expert enough to write this paper themselves.
> >
> > Our difference of opinion is largely philosophical, and partially from my preference to write a complete review (rather than leaving sections blank). When I write, for example, that the methodology "seems quite reasonable" it is only partially a strength. My other meaning is that something has gotten lost between your keyboard (writing the paper) and my keyboard (writing the review) that makes me lack the confidence to write a stronger sentence.
> >
> > I suspect that our philosophical disagreement may lie in whether it is the author's job to write a paper as clearly as possible or the readers job to be smart/hardworking enough to understand it. In writing my review I went back to Ariel's 2009 paper to try to answer the question "is it possible to write something with a similar amount of depth in a clear and understandable manner?" I found that paper much more readable which suggests to me that this paper could (and, therefore, should) be more readable as well. If I were to run into this paper as a reviewer again, I would give a higher score if concepts and results were explained in a more clear manner but, for now, I maintain my current score.
> >
> >
> > To be clear for meta-reviewers deciding what factors to prioritize: My score is largely based on the clarity of the paper which has prevented me from reviewing the paper in sufficient depth needed for a higher score. Not as a result of any known technical issues.

---

> > > ### Author Response · Authors · 2024-08-12
> > >
> > > Thank you for your response. We would like to solicit any specific changes/additions you can suggest to improve the clarity of the paper, and would be happy to incorporate these. Such feedback would be very valuable to us, since we strongly believe in the interdisciplinary value of our work, and are committed to making it accessible to a broader audience.
> > >
> > > Based on the other reviewers' suggestions, we believe the following changes would make the paper easier to read:
> > >
> > > - Additional clarifications for all theorems and lemmas: While it would be difficult to incorporate proofs within the NeurIPS page limit, we can definitely add more intuition for our results. We point to our rebuttal to reviewer 5Qbq as an example, where we give intuitions behind Theorem 3.2. We will rewrite our current exposition to include these additional points.
> > >
> > > - Improved motivation for experiments: Our rebuttal to reviewer 7Nn5 contains further motivation for our experiments, and we plan to add it to the paper to better contextualize our experiment plan.
> > >
> > > - As reviewer 1QjM has suggested, we will add a primer on key learning-theoretic concepts like VC dimension, Rademacher complexity, and pseudo-dimension.
> > >
> > > Finally, while we appreciate the comparison with Procaccia et al. (2009), we note that it’s a journal paper that enjoys unlimited space, so it may set an impossibly high bar for a 9-page conference paper.

---

> > > > ### Comment · Reviewer_W5KA · 2024-08-13
> > > >
> > > > Yes, those sorts of things are exactly what I would hope to see in a very well written paper. In my opinion, a paper is most accessible if it explains ideas in multiple ways (e.g. providing an intuitive outline for a first reading/a non-expert and providing theory/proofs for deeper reading/the more mathematically inclined).
> > > >
> > > > Your final note touches on something I mentioned in my initial review. It may be that fitting all of this content into one paper while explaining it clearly is simply not possible. I agree that it may not be possible in the confines of a conference paper, hence my comment about multiple papers or a journal paper. My rejection suggestion should certainly not be taken as an indictment of the underlying work, but rather as (i) more explanation being needed (as discussed), and (ii) the current paper simply not being a good fit for conference publication, in my opinion.

---

### Official Review · Reviewer_7Nn5 · 2024-07-15

**Soundness:** 3
**Presentation:** 3
**Contribution:** 2
**Rating:** 5
**Confidence:** 4

**Summary:**

This work studies learning the social welfare function from a power mean function class.

- They first consider the cardinal social welfare setting, where the data distribution is over the utilities and social welfare values. They provide the upper bounds on the pseudo-dimensions of the function class and then apply them to bound the generalization loss.

- They then consider pairwise preference setting, where the data distribution is over utilities of a pair of actions and the comparison of their social welfare values. They provide bounds on the VC dimension of the function class and then apply them to bound the generalization loss. They further study two noisy settings, iid noise and logistic noise.

- Finally, they conduct experiments to justify their results.

**Strengths:**

This work studies a new problem of learning social welfare function.

The writing is very clear.

They provide both theoretical and empirical analyses.

**Weaknesses:**

The availability of labels in the real world: In the cardinal setting, the label of the data point is the true social welfare value. I am curious if there really exists any such labeled data set. I have the same question in the pairwise comparison setting.

The technical contribution seems to be limited. It looks like the results are derived by applying standard learning theory results. If there are any technical challenges in deriving the results, e.g., Lemmas 3.1 and 4.1, it would be great if the authors could address them. This is also not a new approach but standard ERM. So far, the problem studied in this work looks like a special case of general learning problems and doesn't require any new techniques.

The experimental results for this problem also look consistent with the phenomena in general learning, e.g., loss decreases with decreasing noise and increasing sample size. What is the takeaway information from the experiments?

**Questions:**

see weaknesses.

---

> ### Author Rebuttal · Authors · 2024-08-06
>
> > The availability of labels in the real world: In the cardinal setting, the label of the data point is the true social welfare value. I am curious if there really exists any such labeled data set. I have the same question in the pairwise comparison setting.
>
> Let us start with pairwise comparisons. In our paper, we use data from the work of Lee et al. on food allocation, from which we derive the learned utility functions of participants/stakeholders. Their data also includes food allocation decisions made by a human decision maker (specifically, a "dispatcher"). These allocation decisions provide ordinal information and induce pairwise comparisons: for each of thousands of decisions in the data, the chosen alternative is preferred to each alternative that was not chosen. Consquently, we have the data to learn the $p$-mean welfare function optimized by the human dispatcher in practice. We did not run this experiment as it requires quite a bit of data processing and does not seem to give generalizable insights. Nevertheless, this shows that pairwise comparison data is, in fact, available, and demonstrates why we'd expect to obtain such data more generally.
>
> The cardinal setting is important to study to understand the complexity of the problem, before analyzing the harder pairwise setting. Obtaining cardinal labels is more challenging but not impossible. We're actively working on this through collaborations with domain experts in public health and disaster management. We're designing experimental platforms to elicit cardinal labels from real decision-makers in these domains. In other words, we're actively working to bridge the gap between theory and practice through ongoing collaborations and experimental design efforts. While it's difficult to convey the full scope of our work within the constraints of anonymous review, we're genuinely committed to making this approach practically viable.
>
>
> > The technical contribution seems to be limited. It looks like the results are derived by applying standard learning theory results. If there are any technical challenges in deriving the results, e.g., Lemmas 3.1 and 4.1, it would be great if the authors could address them. This is also not a new approach but standard ERM. So far, the problem studied in this work looks like a special case of general learning problems and doesn't require any new techniques.
>
>
> We disagree with this comment. While the VC and Rademacher based bounds are standard, deriving these quantities for the specific class of social welfare functions is technically challenging as we outline below. Similarly, while ERM is standard, we demonstrate how to leverage properties such as quasi-convexity of the class of social welfare functions to implement the computationally challenging ERM in practice and enable learning in real world settings. We highlight the novel technical challenges and contributions in our work below.
>
> Lemma 4.1 presented significant technical challenges. For the known weights case (4.1a), a key difficulty was bounding the number of roots of the difference of two weighted power mean functions. This required applying a little-known result by Jameson on counting zeros of generalized polynomials, leading to a tight VC dimension bound of $O(\log d)$, which scales non-trivially with d. For unknown weights (4.1b), we developed a novel proof technique involving analysis of linear dependence among vectors defined by utility differences. This required intricate combinatorial arguments to establish an $O(d \log d)$ upper bound, which isn't derivable from standard learning theory results.
>
> For the lower bound with known weights, we provided a lower bound (4.1c) using a Gray code construction inspired by Bhat and Savage, demonstrating that our upper bound in this case is tight.
>
> Beyond our lemmas, we proved quasi-convexity of our functions with respect to weights for fixed p. This property holds for individual samples but may not extend to empirical loss over multiple samples, since the empirical losses we consider involve a mean over multiple samples. Nevertheless, this observation allowed us to design a practical, performant algorithm, demonstrating real-world applicability while highlighting our problem's complexity.
>
> > The experimental results for this problem also look consistent with the phenomena in general learning, e.g., loss decreases with decreasing noise and increasing sample size. What is the takeaway information from the experiments?
>
> Our experiments offer insights beyond typical learning phenomena, specifically for learning social welfare functions. They validate our hypothesis that gradient descent with fixed $p$ is effective, despite the absence of guaranteed quasi-convexity.
>
> Using semi-synthetic data based on real-world utility vectors, we demonstrate our approach's practicality. Despite the non-convexity in $(\mathbf{w}, p)$, our algorithm consistently converges to parameters close to $(\mathbf{w}^*, p^*)$, which is not captured by our theoretical results. This convergence is crucial, as it indicates that the learned $(\hat{\mathbf{w}}, \hat{p})$ accurately captures the true weights and fairness notion with sufficient samples.
>
>  Key takeaways include:
> 1. Empirical validation of the scaling laws established in our theoretical bounds
> 2. Demonstration of effective learning with realistic sample sizes using a computationally feasible algorithm
> 4. Evidence that learned parameters closely reflect true individual weights and fairness notions (e.g., KL divergence between true and learned weights decreases to <0.1 with sufficient samples)
> 4. Quantification of sample complexity increase with noise under various noise models.
>
>  These results not only support our theoretical contributions but also highlight the practical applicability of our approach in learning and interpreting decision-makers' implicit social welfare functions.

---

> > ### Comment · Reviewer_7Nn5 · 2024-08-13
> >
> > Thanks for the response! My concerns are addressed and I'm happy to increase my rating. I suggest the authors to include the discussion for the first two questions in the updated version.

---

> > > ### Author Response · Authors · 2024-08-13
> > >
> > > Thank you very much, we'd be glad to follow your suggestion.

---

### Official Review · Reviewer_5Qbq · 2024-07-24

**Soundness:** 3
**Presentation:** 3
**Contribution:** 3
**Rating:** 7
**Confidence:** 2

**Summary:**

The paper is about learning social welfare function which belongs to a well-studied family of weighted power mean function, as a way to understand a policy maker's rationale. In particular, the paper focuses on two settings: 1) when the input is the vector of utilities/social welfare, and 2) when the input is a pairwise comparison. The paper derives theoretical bounds for different social welfare information for different kinds of loss function with both known and unknown weights, and

**Strengths:**

The paper focuses on an interesting and important question on social welfare function learning, which has great potentials in social learning and policy making. Overall, it is well-organized and well-presented. The theoretical results are solid and elegant. Overall, the authors are careful and transparent about evaluating both the strengths and weaknesses of their work. The claims/arguments are explored in sufficient depth.

**Weaknesses:**

I find some of the results hard to interpret, for example, the bounds in Theorem 3.2. It would be great if the authors could add intuitions behind the result and better demonstrate the influence of each term.

**Questions:**

1. How should I understand the "pseudo-dimensions" of $M_{w, d}$ and $M_d$ in line 161?

2. For the pairwise comparison setting, does it require access to the pairwise comparisons between all pairs of actions? Can it be extended to a setting where there's only partial comparison available, if so, how does it change the theoretical bound?

**Limitations:**

See above.

---

> ### Author Rebuttal · Authors · 2024-08-06
>
> > I find some of the results hard to interpret, for example, the bounds in Theorem 3.2. It would be great if the authors could add intuitions behind the result and better demonstrate the influence of each term.
>
> We appreciate the reviewer's comments and will add further intuition about the role of each term in our results to make the bounds more interpretable.
>
> Our bounds provide a characterization of the sample complexity of learning social welfare functions. Theorem 3.2, for example, provides bound for learning from cardinal utility data.
>
> The first term in both bounds of Theorem 3.2 depends on $\xi = u_{\max}(u_{\max} - u_{\min})$. This quadratic dependence on the range of utility values is expected, since the $\ell_2$ loss also scales quadratically with the magnitude of the utilities (equivalently with $\xi$).
>
> When the weights $\mathbf{w}$ are known, there is no dependence on $d$ in the first term. However, when $\mathbf{w}$ is unknown, the first term grows as $d\log d$. This means that to keep the first term roughly constant, the number of samples $n$ needs to increase proportionally to $d\log d$ as $d$ grows. This is intuitive since learning the $d$ parameters in the unknown weights case requires a proportional increase in the number of samples.
>
> The second term in both bounds is an artifact of applying the Rademacher bound.
>
> > How should I understand the "pseudo-dimensions" of $M_{w,d}$ and $M_{d}$ in line 161?
>
> The proof of Lemma 3.1 in the appendix provides the formal definitions related to pseudo-dimensions (Definition A.4 will be added for completeness). Intuitively, the pseudo-dimension plays a role similar to the VC-dimension for binary function classes, and these complexity measures directly translate into sample complexity bounds for PAC learning. We will add a short clarifying remark to this effect in Section 3.
>
> > For the pairwise comparison setting, does it require access to the pairwise comparisons between all pairs of actions? Can it be extended to a setting where there's only partial comparison available, if so, how does it change the theoretical bound?
>
> Our results hold even if we only observe comparisons for a polynomial in $d$ number of pairs for known and unknown weights respectively, as long as these pairs are drawn i.i.d. from some underlying distribution. In other words, we do not require access to pairwise comparisons between all pairs.
>
> [1] Anthony M, Bartlett PL. The Pseudo-Dimension and Fat-Shattering Dimension. In: Neural Network Learning: Theoretical Foundations. Cambridge University Press; 1999:151-164.

---

> > ### Comment · Reviewer_5Qbq · 2024-08-12
> >
> > I've carefully read the rebuttal by the authors, my evaluation of the paper remains the same.

---

### Decision · Program_Chairs · 2024-09-25

**Decision:**

Accept (spotlight)

**Comment:**

The paper spots two accepts (7) and an additional positive review.

The single borderline reject review on the paper mentions that there are writing issues, that the paper's contributions don't fit into a single ML paper, and that it should be submitted as an expanded version at a journal instead. Given that the reviewer has not provided specific/precise evidence of writing issues in the paper to the authors nor to me, that there seems to be no real issue with the content itself, and given that having more content than usual for a ML paper should really not be a reason for rejection, I do think the paper should be accepted.

With two positive reviewers at 7 and relatively few weaknesses, I think the paper could be considered for a spotlight. I am marking my confidence as "less certain" as I believe the paper lies somewhere between accept (poster) and accept (spotlight)